# Sampling with Trustworthy Constraints:
# A Variational Gradient Framework

**Xingchao Liu**[*]
Department of Computer Science
University of Texas at Austin
`xcliu@utexas.edu`

**Xin T. Tong**[*]
Department of Mathematics
National University of Singapore
`mattxin@nus.edu.sg`

**Qiang Liu**
Department of Computer Science
University of Texas at Austin
`lqiang@cs.texas.edu`

## Abstract

Sampling-based inference and learning techniques, especially Bayesian inference, provide an essential approach to handling uncertainty in machine learning (ML). As these techniques are increasingly used in daily life, it becomes essential to safeguard the ML systems with various trustworthy-related constraints, such as fairness, safety, interpretability. Mathematically, enforcing these constraints in probabilistic inference can be cast into sampling from intractable distributions subject to general nonlinear constraints, for which practical efficient algorithms are still largely missing. In this work, we propose a family of constrained sampling algorithms which generalize Langevin Dynamics (LD) and Stein Variational Gradient Descent (SVGD) to incorporate a moment constraint specified by a general nonlinear function. By exploiting the gradient flow structure of LD and SVGD, we derive two types of algorithms for handling constraints, including a primal-dual gradient approach and the constraint controlled gradient descent approach. We investigate the continuous-time mean-field limit of these algorithms and show that they have $O(1/t)$ convergence under mild conditions. Moreover, the LD variant converges linearly assuming that a log Sobolev like inequality holds. Various numerical experiments are conducted to demonstrate the efficiency of our algorithms in trustworthy settings.

## 1 Introduction

Efficient approximation and sampling methods of intractable distributions plays a key role in probabilistic machine learning, especially Bayesian inference. Traditionally, this problem has been framed as Monte Carlo sampling. Recently, variational optimization ideas, which frame the sampling problem into functional minimization problem of KL divergence in the space of distributions, have been popularized, as exemplified by Langevin dynamics (through the lens of Wasserstein gradient flow (Jordan et al., 1998)), and Stein variational gradient descent (Liu & Wang, 2016; Liu, 2017; Liu & Wang, 2018). Let $p_0^*$ be a probability density function supported on $\mathbb{R}^d$ from which it is intractable to draw samples exactly; $p_0^*$ can be the posterior distribution in Bayesian inference, or a probabilistic graphical model constructed based on data and expert knowledge. The variational methods consider

---

[*]: Equal contribution. Code is available at `https://github.com/gnobitab/ConstrainedSampling`.

to minimize the following KL divergence objective in the space of distributions, denoted by $P$,

$$\min_{q \in \mathcal{P}} \mathrm{KL}(q \,||\, p_0^*), \tag{1}$$

which, if solved exactly, recovers the target distribution $q = p_0^*$. Langevin dynamics and SVGD can be viewed as solving (1) following two different types of gradient flow under different metrization of $\mathcal{P}$; in practice, both Langevin dynamics and SVGD yield a particle approximation, a set of points (a.k.a. particles) $\{\theta_i\}_{i=1}^n \subset \mathbb{R}^d$, whose empirical measure $q = \sum_{i=1}^n \delta_{\theta_i}/n$ approximates $p_0^*$ in weak convergence; the difference is that Langevin dynamics obtains the particles with a random diffusion process, while SVGD evolves the particles with a deterministic repulsive interacting particle system.

However, most existing works focus on sampling from unconstrained domains, reflecting the fact that Problem (1) is an unconstrainted optimization on $\mathcal{P}$. In many cases, especially trustworthy machine learning, we would like to impose a constraint of high practical importance in addition to approximating $p_0^*$, yielding a constrained variant of (1)

$$\min_{q \in \mathcal{P}} \mathrm{KL}(q \,||\, p_0^*), \quad s.t. \quad \mathbb{E}_q[g(\theta)] \leq 0, \tag{2}$$

where we find the best approximation of $p_0^*$ subject to a moment constraint $\mathbb{E}_q[g(\theta)] \leq 0$, with $g$ a general nonlinear function specified by the users. If $p_0^*$ is the posterior distribution in Bayesian inference, Problem (2) can be viewed as a form of *posterior regularization* (Zhu et al., 2014). Our framework provides a powerful tool for enabling reliable trustworthy machine learning with fairness, safety, interpretability, and other constraints, which we illustrate as follows.

**Fairness Constraints** A motivating example is *Bayesian fairness* (Chakraborty, 2020; Ji et al., 2020; Dimitrakakis et al., 2019), in which, in addition to approximating the posterior distribution, we want to enforce equalized odds or other fairness constraints on different demographic subgroups.

Specifically, suppose we are given a dataset $\mathcal{D} = \{x^{(i)}, y^{(i)}, z^{(i)}\}_{i=1}^N$ consisting of the feature vector $x^{(i)}$, a label $y^{(i)}$, and a protected attribute $z^{(i)}$ (e.g., male vs. female), and we want to fit it with a prediction model $\hat{y}(x; \theta)$ described by a parameter $\theta$. In typical Bayesian inference, we are interested in finding a distribution $q$ on $\theta$ to approximate the posterior distribution $p_0^*(\theta) = p(\theta \mid \mathcal{D})$. In fair Bayesian inference, we also hope $q$ to satisfy certain fairness constraints. For example, to control the disparate impact, one can introduce a constraint of $\mathbb{E}_q[g(\theta)] \leq 0$ with

$$g(\theta) = \ell_{fair}(\theta) - \epsilon, \qquad \ell_{fair}(\theta) = (\mathrm{cov}_{\mathcal{D}}[z, \; \hat{y}(x; \theta)])^2, \tag{3}$$

where we enforce the prediction $\hat{y}(x; \theta)$ to be uncorrelated with the protected attribute $z$. Therefore, solving (2) allows us to obtain the best approximation of $p_0^*$ subject to the fairness constraint.

**Safety and Interpretability Constraints** It is difficult to control and make sense of the behavior of large AI models such as deep neural networks. In many applications, such as healthcare, robotics, and AI-based systems, it is essential to add guardrails to ensure that the AI models stay within a pre-specified safety region. For example, we may want to ensure that the prediction $\hat{y}(x, \theta)$ must fall inside a pre-specified interval $[\hat{y}_{0,-}(x), \hat{y}_{0,+}(x)]$, which can be ensured by $\mathbb{E}_q[g(\theta)] \leq 0$ with

$$g(\theta) = \mathrm{Dist}(\hat{y}(x, \theta), \; [\hat{y}_{0,-}(x), \hat{y}_{0,+}(x)]),$$

where $\mathrm{Dist}(\cdot, \; \cdot)$ is a point-to-set distance. Similar approach can also be used to control and increase the interpretability of deep neural networks (DNNs), by enforcing that the prediction of DNNs is close to simple interpretable models such as linear classifiers and logic rules.

**Prior-Agnostic Bayesian Inference** The posterior of typical Bayesian inference depends on both the likelihood $\ell(\theta)$ which represents the fitness on data and a prior distribution (let it be $p_0^*$). A poor choice of prior may heavily influence the posterior and yield poor fitness. One approach to automatically limiting the influence of prior is to find a $q$ that minimizes $\mathrm{KL}(q \,||\, p_0^*)$, subject to the moment constraint with $g(\theta) = -\ell(\theta) + \epsilon$, so that we always have a guaranteed high data fitness (i.e., high $\ell(\theta)$) regardless of the choice of the prior. Note that $p_0^*$ here is the prior, while it is the posterior in the two settings above.

**Our Contribution** We advance the frontier of sampling with constraints in the following ways: i) We propose two general approaches to extend SVGD and Langevin dynamics to the constrained setting (1), including a primal-dual gradient method (see Algorithm 1), and a novel constraint controlled gradient descent method (see Algorithm 2). ii) We develop novel theoretical analysis on both methods for both convex and non-convex settings. iii) We demonstrate the power of our approaches on a variety of tasks related to trustworthy machine learning, including fair Bayesian classification, incorporating logic rules into black-box models, and training monotonic neural networks.

**Overview of the Main Algorithms**

We provide a quick summary of the practical algorithms that we develop for solving (1), so that the readers with a practical interest can skip the main theoretical derivation, which is in Section 2.1.

Our algorithms iteratively update a set of $n$ particles $\{\theta_{i,t}\}_{i=1}^n \subset \mathbb{R}^d$ through iteration $t$, such that its empirical distribution, denoted as $q_t = \sum_{i=1}^n \delta_{\theta_{i,t}}/n$, approximately solves the constrained Problem 2 in a proper sense when $t \to +\infty$ and $n \to +\infty$. The updates of our algorithm for the SVGD and Langevin cases are:

$$\text{SVGD:} \quad \theta_{i,t+1} = \theta_{i,t} + h\mathbb{E}_{\theta \sim q_t}[(\nabla \log p_0^*(\theta) - \lambda_t \nabla g(\theta))k(\theta, \theta_{i,t}) + \nabla_\theta k_t(\theta, \theta_{i,t})], \quad (4)$$

$$\text{Langevin:} \quad \theta_{i,t+1} = \theta_{i,t} + h(\nabla \log p_0^*(\theta_{i,t}) - \lambda_t \nabla g(\theta_{i,t})) + \sqrt{2h}\xi_{i,t}, \quad (5)$$

where $\{\xi_{t,i}\}$ are i.i.d. standard Gaussian noise and $h$ is a step size. The updates modify the standard SVGD and Langevin update rules by introducing an extra $\lambda_t \nabla g(\theta)$ term to account the constraint in (2), where $\lambda_t \geq 0$ serves as a Lagrange multipler, whose update rule is described below.

**Primal-dual Gradient Method**  A straightforward approach is to update $\lambda_t$ by performing projected gradient descent on the dual problem of (2). As we show in Section 2.2, for both the SVGD and Langevin case, this can be achieved by,

$$\lambda_t = \max(\lambda_{t-1} + \tilde{h}\mathbb{E}_{q_t}[g(\theta)], \, 0), \quad (6)$$

where $\tilde{h}$ is a step size that can different from $h$. This is similar to the typical primal-dual gradient method on finite dimensional optimization: we increase $\lambda_{t-1}$ if $\mathbb{E}_{q_t}[g(\theta)] \geq 0$ (constraint violated) and decrease it if the constraint is met. See Algorithm 1, and the detailed version in Algorithm 3 in the appendix.

**Constraint Controlled Gradient Method**  Because $\lambda_t$ is iteratively updated in (14), the result can be sensitive to the initialization $\lambda_0$ and the learning rate $\eta$ of $\lambda$. In Section 2.3, we propose an alternative "constraint controlled" method so (4) and (5) yield the steepest descent on the KL objective while ensuring that the solution converges to the feasible set rapidly when it is violated. The derived update $\lambda_t$ is different for the Langevin and SVGD cases and is shown below:

$$\text{Langevin:} \quad \lambda_t = \max\left(\frac{\alpha\mathbb{E}_{q_t}[g] + \mathbb{E}_{q_t}[(\nabla \log p_0^*)^\top \nabla g + \nabla^\top \nabla g]}{\mathbb{E}_{q_t}[\|\nabla g\|^2]}, \, 0\right), \quad (7)$$

$$\text{SVGD:} \quad \lambda_t = \max\left(\frac{\alpha\mathbb{E}_{\theta \sim q_t}[g(\theta)] + \mathbb{E}_{\theta,\theta' \sim q_t}[\nabla_{\theta'} g(\theta')^\top (\nabla \log p_0^*(\theta) + \nabla_\theta)k_t(\theta, \theta')]}{\mathbb{E}_{\theta,\theta' \sim q_t}[\nabla g(\theta)^\top \nabla g(\theta')k_t(\theta, \theta')]}, \, 0\right), \quad (8)$$

where $\alpha > 0$ is a coefficient. Note that the $\lambda_t$ here is completely decided by the information at the $t$-th iteration and does not need to be iteratively updated from $\lambda_{t-1}$ like (6). See Algorithm 2 and more details in Algorithm 4 in Appendix.

## 2  Main Results

We first introduce the background on SVGD and Langevin through the view of minimizing (1) via functional steepest descent in Section 2.1. We then introduce our two approaches to extending SVGD and Langevin dynamics for solving the constraint optimization in (2) in Section 2.2 and Section 2.3. We provide theoretical analysis of both approaches with and without convexity assumptions.

### 2.1  Review on Langevin Dynamics and SVGD: Sampling with Steepest Descent

We provide a unified introduction to Langevin dynamics and Stein variational gradient descent (SVGD), which can be viewed as minimizing $\mathrm{KL}(q \,||\, p_0^*)$ with two different types of gradient flow on the space of distributions $\mathcal{P}$. Our results apply to both algorithms. For audience with special interest in one of the two algorithms, we recommend ignoring the other one in the first read.

Assume we start from an initial density $q_0$ and the associated random variable $\theta_0 \sim q_0$. We consider moving the random variable $\theta_t$ along a time-dependent vector field $\phi_t : \mathbb{R}^d \to \mathbb{R}^d$. In other words, $\theta_t$ is driven by an ordinary differential equation (ODE): $\mathrm{d}\theta_t/\mathrm{d}t = \phi_t(x_t)$. Let $q_t$ be the distribution

of $\theta_t$ at time $t$, which is known to follow the Fokker-Planck equation $dq_t/dt = -\nabla^\top(\phi_t q_t)$, where $\nabla^\top f(\theta) := \sum_{\ell=1}^d \partial_{\theta^\ell} f^\ell(\theta)$ denotes the divergence operator of a vector-valued function $f \colon \mathbb{R}^d \to \mathbb{R}^d$ with $\theta^\ell$ and $f_\ell$ the $\ell$-th element of $\theta$ and $f$, respectively; here we use the "$\top$" notation because we formally view $\nabla$ as a $d$-dimensional column vector.

To decrease $\mathrm{KL}(q_t \,\|\, p_0^*)$ as fast as possible, we want to choose $\phi_t$, from a pre-decided candidate function set $\mathcal{F}_t$, so that the decreasing rate $-\frac{\mathrm{d}}{\mathrm{d}t}\mathrm{KL}(q_t \,\|\, p_0^*)$ is maximized. One can show that

$$-\frac{\mathrm{d}}{\mathrm{d}t}\mathrm{KL}(q_t \,\|\, p_0^*) = \mathbb{E}_{\theta \sim q_t}\left[ (\nabla \log p_0^*(\theta) - \nabla \log q_t(\theta))^\top \phi_t(\theta) \right] := \mathbf{R}_{q_t, p_0^*}[\phi_t], \qquad (9)$$

which is a linear functional $\mathbf{R}_{q_t, p_0^*}$ acting on $\mathcal{F}_t$. Assume $\mathcal{F}_t$ is a Hilbert space, equipped with an inner product $\langle \cdot, \cdot \rangle_{\mathcal{F}_t}$ and a norm $\|\cdot\|_{\mathcal{F}_t}$. By Riesz representation theorem, assuming $\mathbf{R}_{q_t, p_0^*}$ is a continuous linear functional, there exists an element $\mathbf{r}_{q_t, p_0^*} \in \mathcal{F}_t$, such that $\mathbf{R}_{q_t, p_0^*}[\phi] = \langle \mathbf{r}_{q_t, p_0^*}, \phi \rangle_{\mathcal{F}_t}$. We call $\mathbf{r}_{q_t, p_0^*}$ the Riesz representation of $\mathbf{R}_{q_t, p_0^*}$ in $\mathcal{F}_t$. The optimal choice of $\phi_t$ in $\mathcal{F}_t$ can be framed into an optimization:

$$\phi_t = \arg\min_{\phi \in \mathcal{F}_t}\left\{ -\langle \mathbf{r}_{q_t, p_0^*}, \, \phi \rangle_{\mathcal{F}_t} + \frac{1}{2}\|\phi\|_{\mathcal{F}_t}^2 \right\} = \mathbf{r}_{q_t, p_0^*}, \qquad \mathbb{D}_{\mathcal{F}_t}(q_t, p_0^*)^2 = \left\|\mathbf{r}_{q_t, p_0^*}\right\|^2, \qquad (10)$$

where we add a regularization $\|\phi\|_{\mathcal{F}_t}^2$ to constrain the scale of $\phi$, yielding the optimal solution $\phi_t = \mathbf{r}_{q_t, p_0^*}$; here we defined $\mathbb{D}_{\mathcal{F}_t}(q_t, \, p_0^*)^2$ to be the corresponding maximum descending rate of KL divergence, which can be viewed as a discrepancy measure between $q_t$ and $p_0^*$. In this work, we always assume that $\mathcal{F}_t$ is sufficiently large, so that $\mathbb{D}_{\mathcal{F}_t}(q_t, \, p_0^*) = 0$ implies $q_t = p_0^*$.

Langevin dynamics and SVGD can be viewed as using different Hilbert spaces $\mathcal{F}_t$, hence yielding different Riesz representation $\mathbf{r}_{q_t, p_0^*}$ for the linear operator $\mathbf{R}_{q_t, p_0^*}$.

**Langevin Dynamics**   Taking $\mathcal{F}_t$ to be $\mathcal{L}_{q_t, 2}$, the Hilbert space of $\mathbb{R}^d \to \mathbb{R}^d$ maps equipped with inner product $\langle \phi, \phi' \rangle_{\mathcal{L}_{q_t, 2}} := \mathbb{E}_{\theta \sim q_t}[\phi(\theta)^\top \phi'(\theta)]$. Then it is immediate to see from (9) that

$$\phi_t(\theta) = \mathbf{r}_{q_t, p_0^*}(\theta) = \nabla \log p_0^*(\theta) - \nabla \log q_t(\theta). \qquad (11)$$

In this case, $\mathbb{D}_{\mathcal{F}_t}(q_t, \, p_0^*)^2$ is the Fisher divergence $\mathbb{E}_{\theta \sim q_t}\left[ \|\nabla \log p_0^*(\theta) - \nabla \log q_t(\theta)\|^2 \right]$. Note that the Fokker Planck equation, $dq_t/dt = -\nabla^\top((\nabla \log p_0^* - \nabla \log q_t)q_t) = -\nabla^\top(\nabla \log p_0^*(\theta)q_t(\theta)) + \nabla^\top \nabla q_t(\theta)$, coincides with the density functions of the Langevin diffusion process $d\theta_t = \nabla \log p_0^*(\theta_t)dt + \sqrt{2}dW_t$, whose time-discretization yields the (unadjusted) Langevin Monte Carlo method; here $W_t$ is the standard Wiener process. Meanwhile, other discretization methods also exist, see e.g. Wibisono (2018); Salim et al. (2020).

**Stein Variational Gradient Descent (SVGD)**   In SVGD, we take $\mathcal{F}_t$ to the reproducing kernel Hilbert space of a positive definite and continuously differentiable kernel $k_t(\theta, \theta')$. By the reproducing property of RKHS and *Stein identity* (see Liu & Wang (2016)), one can show that the optimal $\phi_t$ is

$$\phi_t(\cdot) = \mathbf{r}_{q_t, p_0^*}(\cdot) = \mathbb{E}_{\theta \sim q_t}[\nabla \log p_0^*(\theta)k_t(\theta, \cdot) + \nabla_\theta k_t(\theta, \cdot)], \qquad (12)$$

Related, $\mathbb{D}_{\mathcal{F}_t}(q_t, p_0^*)$ reduces to kernel Stein discrepancy (Liu et al., 2016; Chwialkowski et al., 2016):

$$\mathbb{D}_{\mathcal{F}}(q_t, p_0^*)^2 = \left\|\mathbf{r}_{q_t, p_0^*}\right\|_{\mathcal{F}_t}^2 = \mathbb{E}_{\theta \sim q_t}[(\nabla_\theta \log p_0^*(\theta) + \nabla_\theta)^\top (\nabla_{\theta'} \log p_0^*(\theta') + \nabla_{\theta'})k_t(\theta, \theta')].$$

A nice property of the $\phi_t$ in (12), compared with the $\phi_t$ in (11), is that it depends on $q_t$ only through the expectation operator $\mathbb{E}_{q_t}$, which enables a direct particle implementation of $d\theta_t = \phi_t dt$, instead of resolving to a diffusion process. Specifically, if we initialize $q_0 = \sum_{i=1}^n \delta_{\theta_{i,0}}/n$ to be the empirical measure of a set of particles $\{\theta_{i,0}\}_i$, then $q_t$ remains to be an empirical measure, that is, $q_t = \sum_{i=1}^n \delta_{\theta_{i,t}}/n$, where the particles are evolved by

$$\frac{\mathrm{d}}{\mathrm{d}t}\theta_{i,t} = \mathbb{E}_{\theta \sim q_t}[\nabla \log p_0^*(\theta)k_t(\theta, x_{i,t}) + \nabla_\theta k_t(\theta, \theta_{i,t})], \quad \forall i = 1, \ldots, n,$$

This is a set of differential equations coupled by the *mean field* $q_t$. Following Liu & Wang (2016), the bandwidth parameter of the kernel $k_t(\theta, \theta')$ at time $t$ can depend on the current particles $\{\theta_{i,t}\}_i$ via the median trick.

## 2.2 Primal-Dual Gradient Method

Introducing a Lagrange multipler $\lambda$, Problem (2) is equivalent to the following minimax problem:

$$\min_{q \in \mathcal{P}} \max_{\lambda \geq 0} \left\{ L(q, \lambda) = \mathrm{KL}(q \,||\, p_0^*) + \lambda \mathbb{E}_q[g(\theta)] \right\}. \tag{13}$$

We solve the minimax problem by alternatively updating $\lambda$ and $q$ with gradient descent. Note that we can rewrite $L(q, \lambda)$ into $L(q, \lambda) = \mathrm{KL}(q \,||\, p_\lambda^*) + \Phi(\lambda)$, with

$$p_\lambda^*(\theta) = p_0^*(\theta) \exp(-\lambda g(\theta) + \Phi(\lambda)), \quad \text{and} \quad \Phi(\lambda) = \min_{q \in \mathcal{P}} L(q, \lambda) = -\log \mathbb{E}_{\theta \sim p_0^*}[\exp(-\lambda g(\theta))],$$

and $p_\lambda^* = \arg\min_{q \in \mathcal{P}} L(q, \lambda)$. Therefore, if we want to minimize $L(q, \lambda_t)$ with a fixed $\lambda_t$, we should update $q_t$ to move it towards $p_{\lambda_t}^*$, which can be done by following the functional gradient descent with either SVGD or Langevin dynamics: $\mathrm{d}\theta_t/\mathrm{d}t = \mathbf{r}_{q_t, p_{\lambda_t}^*}(\theta_t)$. With a fixed $q$, we perform standard projected gradient ascent on $\lambda$ shown in (6):

$$\lambda_t = \max(\lambda_{t-1} + \tilde{h} \mathbb{E}_{q_t}[g(\theta)], \, 0), \tag{14}$$

where $\tilde{h}$ is a step size. In the continuous time limit ($\tilde{h} \to 0$), we find the $\lambda_t$ follows a projected ordinary differential equation. It can be written jointly with the density $q_t$ of $\theta_t$ as a primal-dual gradient flow (PDGF):

$$\frac{\mathrm{d}q_t}{\mathrm{d}t} = -\nabla \cdot (\phi_t q_t) = -\nabla \cdot (\mathbf{r}_{q_t, p_{\lambda_t}^*} q_t), \quad \frac{\mathrm{d}\lambda_t}{\mathrm{d}t} = [\eta \mathbb{E}_{q_t}[g]]_{\lambda_t, +}, \tag{15}$$

with $[a]_{\lambda, +} = \mathbb{I}(\lambda \leq 0)\max(a, 0) + \mathbb{I}(\lambda > 0)a$ and $\eta = \tilde{h}/h$ characterizes the update speed of $\lambda_t$ relative to $q_t$. Again, $\mathbf{r}_{q_t, p_{\lambda_t}^*}$ is defined by (11) for the Langevin dynamics, and (12) for the SVGD, as long as $p_0^*$ is replaced by $p_{\lambda_t}^*$.

**Numerical Implementation** To implement (15) through a particle-based algorithm, we use Euler discretization and approximate $q_t$ with empirical measure $\sum_{i=1}^n \delta_{\theta_{i,t}}/n$, where $\{\theta_{i,t}\}_{i=1}^n$ are the particles at time $t$. This yields the updates of SVGD and Langevin shown in (4) and (5). Note that in (5), we run $n$ parallel chains of Langevin dynamics, which are coupled since they share the common $\lambda$. The empirical distribution of $\{\theta_{i,t}\}$ is used to approximate $\mathbb{E}_{q_t}[g(\theta)]$ in (14). See Algorithm 1 and more details in Algorithm 3.

**Convergence Analysis**

One problem of primal-dual gradient method is that the algorithm does not try to meet the constraint directly. Instead, it tries to minimize $\mathrm{KL}(q \,||\, p_\lambda^*)$ which involves the constraint. So in order to prove the convergence, we need to assume that $\mathrm{KL}(q \,||\, p_\lambda^*)$ and its maximum descending rate $\mathbb{D}_{\mathcal{F}}(q, p_\lambda^*)$ dominate the constraint function $g$. In particular, we consider the following assumptions:

**Assumption 2.1.** *There exist positive constants $c_1, c_2 < \infty$, such that for any $t \geq 0$,*

$$(\mathbb{E}_{q_t}[g] - \mathbb{E}_{p_{\lambda_t}^*}[g])^2 \leq c_1 \mathbb{D}_{\mathcal{F}_t}(q_t, \, p_{\lambda_t}^*)^2 \tag{16}$$

$$(\mathbb{E}_{q_t}[g] - \mathbb{E}_{p_{\lambda_t}^*}[g])^2 \leq c_2 \mathrm{KL}(q_t \,||\, p_{\lambda_t}^*). \tag{17}$$

| **Algorithm 1** Primal-Dual Method | **Algorithm 2** Constraint Controlled Method |
|---|---|
| Initialize the particles $\{\theta_{i,0}\}_{i=1}^n$ and $\lambda_0$. Let $q_t = \sum_i \delta_{\theta_{i,t}}/n$.
**for** iteration $t$ **do**
  Update the particles $\{\theta_{i,t}\}_{i=1}^n$ by Eq. (4) (SVGD), or Eq. (5) (Langevin).
  Update $\lambda_t$ by Eq. (14).
**end for** | Initialize the particles $\{\theta_{i,0}\}_{i=1}^n$. Let $q_t = \sum_i \delta_{\theta_{i,t}}/n$.
**for** iteration $t$ **do**
  **If** SVGD, update $\{\theta_{i,t}\}_{i=1}^n$ by Eq. (4), and then update $\lambda_t$ by Eq. (8)
  **If** Langevin, update $\{\theta_{i,t}\}_{i=1}^n$ Eq. (5), and then Update $\lambda_t$ by Eq. (7).
**end for** |

Condition (17) always holds when $g$ is a bounded function, due to the Pinsker's inequality (See e.g., Lemma A.1 of Cui & Tong (2021)). Condition (16) holds if $g$ can be written into a form of $g(\theta) = a + \nabla \log p_\lambda^*(\theta)^\top \psi(\theta) + \nabla^\top \psi(\theta)$ for some $a \in \mathbb{R}$, $\psi \in \mathcal{F}_t$ and $\|\psi\|_{\mathcal{F}_t}^2 \leq c_1$ (see Lemma B.1 in Appendix). It can also be obtained by (17) and a log-Sobolev like inequality (18), which we will discuss later.

**Assumption 2.2.** *There exists $0 < v_0 \leq v_1 < \infty$ such that $v_0 \leq \mathrm{var}_{\theta \sim p^*_{\lambda_t}}[g(\theta)] \leq v_1$ for $\forall t \geq 0$.*

This is a mild assumption, and holds when $g$ is bounded and is not a constant on the support of $p^*_{\lambda_t}$.

Our analysis is based on the following Lyapunov function for the minimax problem on $L(q, \lambda)$:

$$E(q, \lambda) = L(q, \lambda) - 2\Phi(\lambda) + \max_{\lambda \geq 0} \Phi(\lambda).$$

**Lemma 2.3.** *We have $E(q, \lambda) \geq 0$, and $E(q, \lambda) = 0$ iff $(q, \lambda)$ is a saddle point of $L(q, \lambda)$.*

Under mild conditions, we can show $E(q_t, \lambda_t)$ is decaying along PDGF. Moreover, we can find a solution that meets the KKT requirement approximately.

**Theorem 2.4.** *If (16) holds and $0 < \eta < 1/(2c_1)$, then $E(q_t, \lambda_t)$ decreases monotonically following the primal-dual gradient flow in (15),*

$$-\frac{\mathrm{d}}{\mathrm{d}t}E(q_t, \lambda_t) \geq \Delta(q_t, \lambda_t) := (1 - 2c_1\eta)\mathbb{D}_{\mathcal{F}_t}(q_t, p^*_{\lambda_t})^2 + \frac{1}{2}\eta(\mathbb{E}_{q_t}[g])^2 \times \mathbb{I}(\lambda > 0 \ \text{or} \ \mathbb{E}_{q_t}[g] > 0).$$

*Also (15) finds an approximate solution with an $O(1/T)$ rate:* $\min_{t \in [0, T]} \Delta(q_t, \lambda_t) \leq \frac{1}{T}E(q_0, \lambda_0), \forall T \geq 0$.

Note that $\Delta(q, \lambda)$ is a measure of optimality. This is because $\Delta(q, \lambda) = 0$ implies the KKT condition of the constrained optimization (1) holds, which is $q = p^*_\lambda$, $\mathbb{E}_q[g] \leq 0$ and $\lambda\mathbb{E}_q[g] = 0$.

**Linear Convergence with Log-Sobolev like Condition** We can further show a linear convergence of KL divergence to the optimal solution by assuming a Logarithmic Sobolev like inequality.

**Assumption 2.5.** *There exists a positive constant $c_3 < \infty$, such that for any $t \geq 0$,*

$$\mathrm{KL}(q_t \,||\, p^*_{\lambda_t}) \leq c_3 \mathbb{D}_{\mathcal{F}_t}(q_t, \, p^*_{\lambda_t})^2. \tag{18}$$

**Theorem 2.6.** *Suppose problem (13) has a solution $(q^*, \lambda^*)$, and Assumptions 2.1,2.2 and 2.5 hold. The PDGF following (15) with $0 < \eta < 1/(2c_1)$ will converge to $q^*$ linearly in $\mathrm{KL}$ divergence:*

$$\mathrm{KL}(q_t||q^*) \leq \alpha_1 E(q_t, \lambda_t) \leq \alpha_1 \exp(-\alpha_2 t)E(q_0, \lambda_0),$$

*where $\alpha_1 = \max\{1 + \frac{1}{2}c_2, \frac{v_1}{2v_0} + \frac{1}{2v_0}\}$ and $\alpha_2 = \min\{\frac{1}{c_3}(1 - c_1\eta), \frac{1}{2}\eta v_0\}$.*

Assumption 2.5 is equivalent to the log-Sobolev inequality in the Langevin case (when $\mathcal{F}_t = \mathcal{L}_{q_t, 2}$ and $\mathbb{D}_{\mathcal{F}}(\cdot, \cdot)^2$ is Fisher divergence), which holds when $\log p^*_\lambda$ is a bounded perturbation of a strongly concave function (Holley & Stroock, 1986). In particular, it holds when $p^*_0$ is strongly log-concave and $g$ is convex. So it can be seen as a strong convexity (concavity) assumption. In the optimization literature, e.g. Nesterov et al. (2018), it is well known such assumption is necessary for linear convergence; without it, we can usually only show $O(1/t)$ convergence similar to Theorem 2.4.

For SVGD, Assumption 2.5 is less well understood. If one replaces $q_t$ with any probability measure $q$, then (18) would fail to hold for kernel Stein discrepancy (KSD) because if $q_t$ is a discrete particle measure, we would have $\mathrm{KL}(q_t \,||\, p^*_\lambda) = \infty$ but $\mathbb{D}_{\mathcal{F}_t}(q_t, p^*_\lambda) < \infty$ under some mild conditions on the kernel $k_t$; see also Lemma 36 of Duncan et al. (2019). On the other hand, it is unclear if (17) will hold for the smaller class of densities where $\{q_t \colon t \in [0, \infty)\}$ takes place in SVGD given that the initialization $q_0$ is sufficiently regular (and in what sense). Therefore, Theorem 2.6 is not readily applicable to SVGD. We hope future works can draw more understandings on this issue.

## 2.3 Constraint Controlled Gradient Descent

The results of the primal-dual gradient method can be sensitive to the initialization $\lambda_0$ and the learning rate $\eta$ of $\lambda$. We now propose a "constraint controlled" method, which finds a constrained variant of steepest descent direction $\phi_t$ that yields the steepest descent on the KL objective like Section 2.1, while ensuring that the solution converges to the feasible set rapidly when it is violated.

Following Section 2.1, assume we update $q_t$ by $\mathrm{d}\theta_t = \phi_t(\theta_t)\mathrm{d}t$, and want to decide the optimal $\phi_t$. To solve the constrained optimization (2), in addition to maximizing the descending rate of KL

divergence as Section 2.1, we also want to ensure that the constraint $\mathbb{E}_{q_t}[g]$ is properly controlled, such that 1) if the constraint is not met (i.e., $\mathbb{E}_{q_t}[g] \geq 0$), we should monotonically decrease $\mathbb{E}_{q_t}[g]$, and 2) after the constraint is met (i.e., $\mathbb{E}_{q_t}[g] \leq 0$), we should monotonically descend the loss $\mathrm{KL}(q_t, p_0^*)$, while ensuring that the solution stays within the constraint set. Note that

$$\frac{\mathrm{d}}{\mathrm{d}t} \mathbb{E}_{q_t}[g] = \mathbb{E}_{q_t}[\nabla g(\theta)^\top \phi_t(\theta)] := \mathbf{S}_{q_t,g}[\phi_t] = \langle \mathbf{s}_{q_t,g}, \ \phi_t \rangle_{\mathcal{F}_t},$$

where $\mathbf{S}_{q_t,g}$ is a linear operator on $\mathcal{F}_t$, whose Riesz representation is assumed to be $\mathbf{s}_{q_t,g}$. Generalizing the functional steepest descent idea in (10), we propose to set $\phi_t$ to be the solution of

$$\phi_t = \underset{\phi \in \mathcal{F}_t}{\arg\min} \left\{ -\langle \mathbf{r}_{q_t,p_0^*}, \ \phi \rangle_{\mathcal{F}_t} + \frac{1}{2} \|\phi\|_{\mathcal{F}_t}^2 \quad s.t. \quad \langle \mathbf{s}_{q_t,g}, \ \phi \rangle_{\mathcal{F}_t} \leq -\alpha \mathbb{E}_{q_t}[g] \right\}, \quad (19)$$

where $\alpha \geq 0$ is a control coefficient. This ensures that, if the constraint is not met (i.e. $\mathbb{E}_{q_t}[g] > 0$), the constraint is descending ($\frac{\mathrm{d}}{\mathrm{d}t} \mathbb{E}_{q_t}[g] \leq -\alpha \mathbb{E}_{q_t}[g] < 0$). If the constraint is met ($\mathbb{E}_{q_t}[g] \leq 0$), then it allows $\mathbb{E}_{q_t}[g]$ to increase, but with a rate smaller than $-\alpha \mathbb{E}_{q_t}[g]$, which decreases towards zero when the solution approaches the constraint boundary $\{q : \mathbb{E}_q[g] = 0\}$; this on one hand allows us to have the flexibility to choose $\phi$ to decrease $\mathrm{KL}(q \,\|\, p_0^*)$), and on the other hand ensures the solution is confined inside the constraint set. The high-level idea here is similar to the control barrier functions in control theory (e.g., Ames et al., 2019), which works in completely different settings. Note that the constraint in (19) can be viewed as a linearization of the constant $\mathbb{E}_q[g] \leq 0$ around $q_t$, as $\alpha \mathbb{E}_{q_{t+1/\alpha}}[g] = \alpha \mathbb{E}_{q_t}[g] + \frac{\mathrm{d}}{\mathrm{d}t} \mathbb{E}_{q_t}[g] = \alpha \mathbb{E}_{q_t}[g] + \langle \mathbf{s}_{q_t,g}, \ \phi \rangle_{\mathcal{F}_t}$, while the loss in (19) can be viewed as a simple quadratic approximation of $\mathrm{KL}(q \,\|\, p_0^*)$. Therefore, (19) can be viewed as a functional variant of sequential quadratic programming (Nocedal & Wright, 2006).

Using the Lagrange duality of (19), it is easy to derive that $\phi_t = \mathbf{r}_{q_t,p_0^*} - \lambda_t \mathbf{s}_{q_t,g} = \mathbf{r}_{q_t,p_{\lambda_t}^*}$, with

$$\lambda_t = \max \left( \frac{\alpha \mathbb{E}_{q_t}[g] + \langle \mathbf{r}_{q_t,p_0^*}, \ \mathbf{s}_{q_t,g} \rangle_{\mathcal{F}_t}}{\|\mathbf{s}_{q_t,g}\|_{\mathcal{F}_t}^2}, \ 0 \right). \quad (20)$$

See Lemma B.2 in Appendix. Therefore, the density following the constraint controlled gradient flow (CCGF) should satisfy:

$$\frac{\mathrm{d}q_t}{\mathrm{d}t} = -\nabla \cdot (\phi_t q_t) = -\nabla \cdot (\mathbf{r}_{q_t,p_{\lambda_t}^*} q_t). \quad (21)$$

This is similar to PDGF (15) as $q_t$ is driven towards $p_{\lambda_t}^*$. But the $\lambda_t$ in PDGF is recursively updated by the algorithm, while CCGF has the explicit formula for $\lambda_t$. Intuitively, CCGF will be more efficient.

Next, we obtain Langevin and SVGD variants by taking $\mathcal{F}_t$ to be $\mathcal{L}_{q_t,2}$ and RKHS, respectively.

**Langevin Dynamics and its implementation** Assume $\mathcal{F}_t = \mathcal{L}_{q_t,2}$. We have $\mathbf{r}_{q_t,p_0^*} = \nabla \log p_0^* - \nabla \log q_t$ and $\mathbf{s}_{q_t,g} = \nabla g$. Hence $\phi_t = \nabla \log p_0^* - \lambda_t \nabla g - \nabla \log q_t$, with $\lambda_t$ defined in (7). The evolution of $q_t$ associated with $\mathrm{d}\theta = \phi_t(\theta)\mathrm{d}t$ is standard Fokker Planck equation associated with $p_{\lambda_t}^*$, which can be realized by the same parallel Langevin particle dynamics in (5), when setting $q_t = \sum_{i=1}^n \delta_{i,t}/n$. Note that $\lambda_t$ depends on $q_t$ only through the expectation $\mathbb{E}_{q_t}[\cdot]$, which can be replaced by the empirical average of the particles. See Algorithm 2 and more details in Algorithm 4.

**The SVGD Case** Let $\mathcal{F}_t$ be the RKHS of kernel $k_t(\theta, \theta')$. Then it is easy to show that $\mathbf{s}_{q_t,g}(\cdot) = \mathbb{E}_{\theta \sim q_t}[\nabla_\theta g(\theta) k_t(\theta, \cdot)]$. Therefore, $\phi_t(\cdot) = \mathbb{E}_{\theta \sim q_t}[(\nabla \log p_0^*(\theta) - \lambda_t \nabla g(\theta)) k_t(\theta, \cdot) + \nabla_\theta k_t(\theta, \cdot)]$, with $\lambda_t$ defined in (8). Similar to regular SVGD, we set $q_t = \sum_{i=1}^n \delta_{\theta_{i,t}}/n$, which is iteratively updated with SVGD updates with $p_{\lambda_t}^*$ as the target distribution as (4). Also, $\lambda_t$ again only depends on $\mathbb{E}_{q_t}[\cdot]$, which should be evaluated with the empirical mean of the particles. See Algorithm 2 and the detailed version in Algorithm 4.

## Convergence Analysis

Since CCGF tries to meet the constraint explicitly, we have the following attractive properties:

**Theorem 2.7.** *Suppose* $\|\mathbf{s}_{q_t,g}\|_{\mathcal{F}_t} \neq 0$ *in the CCGF* (21)*. Then with* $\phi_t$ *defined in* (20)*, we have*

$$\mathbb{E}_{q_t}[g] \leq \exp(-\alpha t) \mathbb{E}_{q_0}[g], \quad \forall t \geq 0.$$

*If* $\mathbb{E}_{q_{t_0}}[g] \leq 0$ *for some* $t_0 \geq 0$*, then* $\mathbb{E}_{q_t}[g] \leq 0$ *and* $\frac{d}{dt}\mathrm{KL}(q_t \,\|\, p_0^*) \leq 0$ *for all* $t \geq s$*.*

Note that the requirement $\|\mathbf{s}_{q_t,g}\|_{\mathcal{F}_t} \neq 0$ is necessary for the $\lambda_t$ in CCGF to be well defined. If fact, if $\|\mathbf{s}_{q_t,g}\|_{\mathcal{F}_t} = 0$, then $\frac{d}{dt}\mathbb{E}_{q_t}[g] = 0$ along any update direction $\phi$. And if $\mathcal{F}_t = \mathcal{L}_{q_t,2}$, one can show $g$ is a constant function $q_t$-a.s.. In other words, this is an ill-posed scenario where local methods like CCGF have no chance to solve. An assumption slightly stronger than $\lambda_t \neq \infty$ would be assuming it is bounded from above:

**Assumption 2.8.** *There exists an upper bound $\lambda_{\max,+} < \infty$, such that for all time $t$ when the constraint is not satisfied (i.e., $\mathbb{E}_{q_t}[g] > 0$), we have $\lambda_t \leq \lambda_{\max,+}$ in (20).*

This is a mild regularity condition, which holds, for example, if $\mathbb{E}_{q_t}[g]$ and $\|\mathbf{r}_{q_t}\|$ are upper bound, and $\|\mathbf{s}_{q_t,g}\| > c_- > 0$. In the following, we show that, under Assumption 2.8, our algorithm meets the KKT condition with a $O(1/t)$ rate. Further, if Assumption 2.5 holds, then $\mathrm{KL}(q_t \,\|\, q^*)$ converges to zero exponentially fast. Note that the conditions here are milder than that of the primal-dual method (which needs Assumption 2.1 and 2.2).

**Theorem 2.9.** *Let $\{q_t\}$ be the density function of $\mathrm{d}\theta_t = \phi_t(\theta)\mathrm{d}t$ with $\phi_t$ defined in (20). Under Assumption 2.8, we have*

$$\min_{t \in [0,T]} \Delta_2(q_t, \lambda_t) := \mathbb{D}_{\mathcal{F}_t}(q_t,\ p^*_{\lambda_t})^2 + \alpha(-\lambda_t \mathbb{E}_{q_t}[g])_+ \leq \frac{1}{T}\left(\mathrm{KL}(q_0, p^*_0) + \lambda_{\max,+}(\mathbb{E}_{q_0}[g])_+\right).$$

Note that, given that $\mathbb{E}_{q_t}[g] \leq 0$ (see Theorem 2.7), $\Delta_2(q_t, \lambda_t) = 0$ implies the rest of KKT condition (i.e., $q_t = p^*_{\lambda_t}$ and $(-\lambda_t \mathbb{E}_{q_t}[g])_+ = 0$).

We establish below the linear convergence of KL divergence under the log-Sobolev condition (18).

**Theorem 2.10.** *Suppose problem (13) has a solution $(q^*, \lambda^*)$, $\|\mathbf{s}_{q_t,g}\|_{\mathcal{F}_t} \neq 0$, and Assumption 2.5 also holds. Then the CCGF following (21) with $\alpha \leq 1/c_3$ will converge to $q^*$ linearly in $\mathrm{KL}$ divergence:*

$$\mathrm{KL}(q_t \,\|\, q^*) \leq e^{-\alpha t}\mathrm{KL}(q_0 \,\|\, q^*), \quad \forall t \in [0, +\infty).$$

Similar to Theorem 2.6, this result is not readily applicable to the SVGD case because (18) can not be verified for SVGD.

## 3 Experiments

**Algorithms and Settings**  We summarize the tested methods and the unconstrained baselines here: `Langevin`: Vanilla Langevin dynamics. We run $n$ parallel chains of Langevin dynamics for sampling; `SVGD`: Vanilla SVGD with codes from Liu & Wang (2016); `Primal-Dual+Langevin`: Langevin dynamics with Primal-dual gradient descent; `Primal-Dual+SVGD`: SVGD with Primal-dual gradient descent; `Control+Langevin`: Langevin dynamics with constraint controlled gradient descent; `Control+SVGD`: SVGD with constraint controlled gradient descent. For the SVGD methods, we follow the configuration in Liu & Wang (2016). We use the RBF kernel with bandwidth chosen by the standard median trick, that is, we use $k_t(\theta, \theta') = \exp(-\|\theta - \theta'\|^2 / w_t^2))$ where the bandwidth $w_t$ is set by $w_t = \mathbf{Median}\{\|\theta_{i,t} - \theta_{j,t}\| : i \neq j\}$ based on the particles $\{\theta_{i,t}\}_{i=1}^n$ at the $t$-th iteration.

As a remark regarding the choice of kernel, Gorham & Mackey (2017) provided a counter-example that suggests that Stein discrepancy with Gaussian RBF kernel may fail to metrize the weak convergence and suggested to use inverse multi-quadratic (IMQ) kernel which does not suffer from the problem. However, this counter-example assumes to use a fixed bandwidth and does not hold for Gaussian RBF kernel equipped with the median trick which can adapt to the scale of the data better. We leave the study of better and adaptive choice of kernels to future works. In our implementation, we adopt the same decaying step size as suggested in Welling & Teh (2011) for both SVGD and Langevin dynamics, where $h_t = h_0(1.0 + t)^{-0.55}$ and $h_0$ is a hyper-parameter. The step size for the Lagrangian multiplier in primal-dual methods is chosen from $\{0.001, 0.01, 0.1, 1, 10, 100, 1000\}$. The hyper-parameters are determined by grid-search to reach the smallest constraint loss in each experiment.

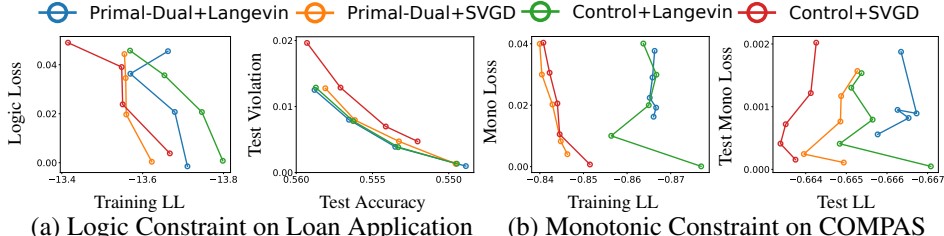

(a) Logic Constraint on Loan Application  (b) Monotonic Constraint on COMPAS

Figure 1: Trade-off curve with different $\epsilon$. 'LL': log-likelihood. 'Mono': monotonicity. Note that the x-axis are flipped so larger values are on the left.

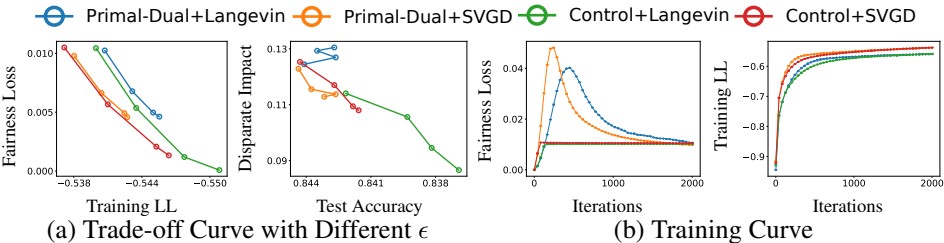

(a) Trade-off Curve with Different $\epsilon$  (b) Training Curve

Figure 2: Experiment results on learning fair Bayesian neural networks. 'LL': log-likelihood. Note that in (a) the x-axis are flipped so larger values are on the left.

**Embedding Logic Rules into Black-box Models** ML models can provide accurate predictions but are difficult to interpret and control explicitly. Our method can be used to enforce the ML model to be consistent with a set of logic rules to improve the interpretability and safety. We illustrate this with a loan classification problem of predicting whether to lend loans to a specific applicant. We impose the two logic rules: (1) an applicant must be denied if she has the lowest credit rank and not employed; (2) an applicant must be approved if she has the highest credit rank and has been employed over 15 years. To apply our method, we set $p_0^*$ to be the typical posterior distribution of Bayesian logistic regression, and define the constraint to $g(\theta) = \ell_{logic}(\theta) - \epsilon$, with $\ell_{logic}(\theta) = \mathbb{E}_{(x,y)\sim\mathcal{D}_{logic}}[\text{Loss}(y, \hat{y}(x; \theta))]$, where $\mathcal{D}_{logic}$ is the uniform distribution on the $(x, y)$ that satisfy the logic constraints, Loss is the classification loss and $\hat{y}(x; \theta)$ is the prediction by the Bayesian logistic regression model. We vary the threshold $\epsilon$ in $\{0.001, 0.01, 0.02, 0.03, 0.04, 0.05\}$ and find the corresponding training log-likelihood (LL). In combination, they are plotted as a trade-off curve shown in in Figure 1(a, left). In Figure 1(a, right), we plot the trade-off curve of the testing accuracy vs. the constraint violation $((\mathbb{E}_{\theta\sim q}[\ell_{logic}(\theta)] - \epsilon)_+)$ on the testing data. We can see that the controlled-based methods tend to enforce the constraints better (reaching lower values in logic loss) in both training and testing set, and the SVGD methods tend to achieve higher test accuracy.

**Training Monotonic Bayesian Neural Networks** In some applications, it can be desirable to enforce the ML prediction to be monotonic w.r.t certain attributes (Karpf, 1991; Sill, 1998). For example, when predicting admission decisions, a fair ML system must admit a student with higher GPA over the students with lower GPA, given that they are identical in the other features. We apply our method to enforce monotonicity in Bayesian neural networks. We use the COMPAS dataset following the setting in Liu et al. (2020). In this case, $p_0^*$ is the posterior of a Bayesian neural network on the data, and $g$ is defined as $g(\theta) = \ell_{mono}(\theta) - \epsilon$ with $\ell_{mono}(\theta) = \mathbb{E}_{x\sim\mathcal{D}}[\|(-\partial_{x_{mono}}\hat{y}(x; \theta))_+\|_1]$. Here, $x_{mono}$ denotes the subset of features to which the output should be monotonic. By varying $\epsilon$ in $\{0.0001, 0.01, 0.1, 0.5\}$, we plot in Figure 1(b) the trade-off curve of LL vs. the monotonic loss on training/testing set, which follow similar trends as Figure 1(a).

**Training Fair Bayesian Neural Networks** We use the constraint defined in (3), and use the *Adult Income* dataset (Kohavi, 1996), which is a classification problem of predicting whether the annual income of a person is $\geq \$50,000$, with gender as the protected attribute. For the experiment, we follow the setting in Martinez et al. (2020); Liu & Vicente (2020). With $\epsilon = \{0.0001, 0.001, 0.005, 0.01\}$, Figure 2(a, left) shows the trade-off curve of training LL vs. fairness loss, and Figure 2(a, right) shows the testing accuracy vs. CV score (Calders & Verwer, 2010), a standard measure of disparate impact. Figure 2(b) shows an example of training curves. We can again see that the control based methods enforce the constraints more strictly and in a faster speed.

**Constrained Sampling on Generative Models** We apply our method on Noise Conditional Score Networks (NCSN) (Song & Ermon, 2019). NCSN models the distribution $p_0^*$ of images by training a neural network to estimate its score function $\nabla \log p_0^*$, then generates images by running Langevin dynamics with the learned score function. Here, we add constraints to draw *rare* samples from the generative model. In this experiment, we constrain the center of the generated images to be a black square, by defining the constraint function to be $g(\theta) = \|\theta_\Omega\| - \epsilon$, where $\theta$ denotes the whole image, $\theta_\Omega$ is a image patch on region $\Omega$, which is the square at the center of image. We use a threshold of $\epsilon = 0.0001$. We use `Control+Langevin` to draw samples under the constraint, as the original NCSN model is based on Langevin dynamics. As shown in Fig. 3, our method successfully generates diverse images without violating the constraints.

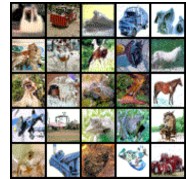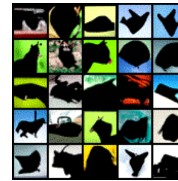

(a) Unconstrained Samples   (b) Control + Langevin

Figure 3: (a): images sampled from the original NCSN model (Song & Ermon, 2019) on CIFAR10. (b): constrained images sampled by `Control+Langevin` using the score function learned by NCSN, where we constrain the center of the generated images to be a black square. Our method generates a variety of *shadow*s following the constraint.

## 4 Related Works, Limitations, Conclusion

**Primal-Dual vs. Constraint-Control** This work has proposed two methods, primal-dual and CCGF. These two methods are derived from two different perspectives. CCGF tries to fit the constraints directly, while the primal-dual operates in a more indirect manner. This is reflected in the proof techniques, as we can directly show the constraint decays linearly for the CCGF method, but we can only control the constrain through a Lyapunov function with the primal dual method. We believe that the conditions required by CCGF is intrinsically weaker than that of primal-dual. This can also be told from the fact that primal-dual method's performance depends on the initialization of $\lambda_t$ and the step size $\eta$. But the CCGF method does not have this issue.

**Related Works** Although sampling on unconstrained domain has been extensively studied, efficient algorithms for constrained settings are largely lacking. A body of works have developed to extend Langevin dynamics and SVGD on Riemannian manifolds (Girolami & Calderhead, 2011; Patterson & Teh, 2013; Liu & Zhu, 2018); these methods, however, rely on computationally efficient characterization of the manifolds and as a result does not work efficiently with constraints specified by general nonlinear inequalities. Projection-based methods (e.g, Sen et al., 2018) can also be developed for sampling on constrained spaces, but again only work for simple constraints with efficient projection maps. Note that our method considers a different setting when the constraint is applied on the population (rather than individual particles) and works for general nonlinear inequality constraints. In terms of problem formulation, our constrained optimization in (2) coincides with the posterior regularization framework of Zhu et al. (2014); but we focus on particle-based inference based on SVGD and Langevin, which was not studied before.

**Limits, Impacts, Future Directions** Our new methods cast a full spectrum of theoretical questions that can not be completed with a single work. In the current work, we mainly focus on the continuous time and the infinite particle limit $n \to +\infty$ when $q_t$ is a smooth probability density. A natural future direction is to study the guarantees with finite particle size $n$ and discrete time. Another valuable direction is to study whether and how linear convergence can be established for SVGD; see Duncan et al. (2019) for an in-depth discussion of this issue for unconstrained SVGD.

On the practical side, since our methods leverage SVGD and Langevin dynamics, they necessarily inherent their limits, such as the sensitivity on the choice of kernel and step sizes. Another practical issue is that we can only enforce constraints on the training set, and hence generalization error need to be considered if we need to strictly enforce constraints on the testing set. Also, this work focuses on a single constraint for simplicity, which is sufficient for most practical applications since different constraints can be combined into a single constraint easily. Extensions to multiple constraints are straightforward but will be studied separately.

In terms of social impact, our method aims to impose trustworthy constraints in Bayesian sampling, and can help increase the reliability, fairness, and interpretability of ML systems applied to daily life. We will release code freely online to promote the applications of our approach, although it does open the possibility of malicious use of our techniques for adversarial purposes.

## Acknowledgements

The work is conducted in part in the statistical learning and AI group in computer science at UT Austin, which is supported in part by CAREER-1846421, SenSE-2037267, EAGER-2041327, and Office of Navy Research, and NSF AI Institute for Foundations of Machine Learning (IFML). Xingchao Liu is supported in part by a funding from BP. X. T. Tong's research is supported by Singapore MOE Academic Research Funds R-146-000-292-114. The authors thank the reviewers for all the suggestions made in the reviewing process.

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
