# A  Details formulations of the Algorithms

---
**Algorithm 3** Primal-Dual Method

---
Initialize the particles $\{\theta_{i,0}\}_{i=1}^n$ and $\lambda_0$.
**for** iteration $t$ **do**
    **If** `Langevin`, update $\theta_{i,t+1} = \theta_{i,t} + h(\nabla \log p_0^*(\theta_{i,t}) - \lambda_t \nabla g(\theta_{i,t})) + \sqrt{2h}\xi_{i,t}$.
    **If** `SVGD`, update

$$\theta_{i,t+1} = \theta_{i,t} + \frac{h}{n}\sum_{j=1}^n [(\nabla \log p_0^*(\theta_{j,t}) - \lambda_t \nabla g(\theta_{j,t}))k_t(\theta_{j,t}, \theta_{i,t})] + \nabla_{\theta_{j,t}} k_t(\theta_{j,t}, \theta_{i,t}).$$

    Update $\lambda_t$ by $\lambda_{t+1} = \max(\lambda_t + \frac{\tilde{h}}{n}\sum_{i=1}^n [g(\theta_{i,t+1})], \, 0)$.
**end for**

---

---
**Algorithm 4** Constraint Controlled Method

---
Initialize the particles $\{\theta_{i,0}\}_{i=1}^n$.
**for** iteration $t$ **do**
    **If** `Langevin`, update

$$\lambda_t = \max\left(\frac{\sum_{j=1}^n \alpha g(\theta_{j,t}) + [(\nabla \log p_0^*(\theta_{j,t}))^\top \nabla g(\theta_{j,t}) + \nabla^\top \nabla g(\theta_{j,t})]}{\sum_{j=1}^n [\|\nabla g(\theta_{j,t})\|^2]}, \, 0\right),$$

    update $\theta_{i,t+1} = \theta_{i,t} + h(\nabla \log p_0^*(\theta_{i,t}) - \lambda_t \nabla g(\theta_{i,t})) + \sqrt{2h}\xi_{i,t}$.
    **If** `SVGD`, update

$$\lambda_t = \max\left(\frac{\sum_{i,j=1}^n \alpha g(\theta_{i,t}) + [\nabla g(\theta_{j,t})^\top (\nabla \log p_0^*(\theta_{i,t}) + \nabla_{\theta_{i,t}})k_t(\theta_{i,t}, \theta_{j,t}])]}{\sum_{i,j=1}^n [\nabla g(\theta_{i,t})^\top \nabla g(\theta_{j,t})k_t(\theta_{i,t}, \theta_{j,t})]}, \, 0\right),$$

    update

$$\theta_{i,t+1} = \theta_{i,t} + \frac{h}{n}\sum_{j=1}^n [(\nabla \log p^*(\theta_{j,t}) - \lambda_t \nabla g(\theta_{j,t}))k_t(\theta_{j,t}, \theta_{i,t}) + \nabla_{\theta_{j,t}} k_t(\theta_{j,t}, \theta_{i,t})].$$

**end for**

---

# B  Proofs

## B.1  Proofs for the primal dual method

**Lemma B.1.** *Suppose $g = a + (\nabla \log p_\lambda^*)^T \psi + \nabla^T \psi$, for some constant $a$ and $\psi \in \mathcal{F}_t$. Then Condition* (16) *holds with $c_1 = \|\psi\|_{\mathcal{F}_t}^2$.*

*Proof.* Note that we can assume $a = 0$ without loss of generality. If $g = (\nabla \log p_\lambda^*)^T \psi + \nabla^T \psi$, by Stein identity $\mathbb{E}_{p_\lambda^*}[g] = 0$. Meanwhile if we let $r = q/p_\lambda^*$

$$\mathbb{E}_q[g] = \int q(\theta)(\nabla \log p_\lambda^*(\theta)^T \psi + \nabla^T \psi(\theta))d\theta$$

$$= \int r(\theta)\nabla^T (p_\lambda^*(\theta)\psi(\theta))d\theta$$

$$= -\int \nabla r(\theta)^T \psi(\theta)p_\lambda^*(\theta)d\theta$$

$$= -\mathbf{R}_{q,p_\lambda^*}(\psi).$$

Therefore

$$(\mathbb{E}_q[g])^2 \leq \langle \mathbf{r}_{q,p_\lambda^*}, \psi \rangle_{\mathcal{F}_t}^2 \leq \mathbb{D}_{\mathcal{F}}(q, p_\lambda^*) \|\psi\|_{\mathcal{F}_t}^2.$$

$\square$

**Lemma 2.3**  *We have $E(q, \lambda) \geq 0$, and $E(q, \lambda) = 0$ iff $(q, \lambda)$ is a saddle point of $L(q, \lambda)$.*

*Proof of Lemma 2.3.*

$$E(q, \lambda) = L(q, \lambda) - 2 \min_{q \in \mathcal{P}} L(q, \lambda) + \max_{\lambda \geq 0} \min_{q \in \mathcal{P}} L(q, \lambda)$$

$$= (L(q, \lambda) - \min_{q \in \mathcal{P}} L(q, \lambda)) + (\max_{\lambda \geq 0} \min_{q \in \mathcal{P}} L(q, \lambda) - \min_{q \in \mathcal{P}} L(q, \lambda)) \geq 0.$$

Therefore, $E(q, \lambda) = 0$ iff $L(q, \lambda) = \min_{q \in \mathcal{P}} L(q, \lambda)$ and $\max_{\lambda \geq 0} \min_{q \in \mathcal{P}} L(q, \lambda) = \min_{q \in \mathcal{P}} L(q, \lambda)$, which implies that $(q, \lambda)$ is a saddle point of $L(q, \lambda)$. $\square$

*Proof of Theorem 2.4.*  If $\lambda_t \geq 0$ or $\mathbb{E}_{q_t}[g] \geq 0$, we have

$$\frac{\mathrm{d}}{\mathrm{d}t} E(q_t, \lambda_t) = -\langle \mathbf{r}_{q_t, p_0^*} - \lambda_t \mathbf{s}_{q_t, g}, \phi_t \rangle_{\mathcal{F}_t} + (\mathbb{E}_{q_t}[g] - 2\mathbb{E}_{p_{\lambda_t}^*}[g]) \frac{\mathrm{d}}{\mathrm{d}t} \lambda_t$$

$$= - \left\| \mathbf{r}_{q_t, p_{\lambda_t}^*} \right\|_{\mathcal{F}_t}^2 + (\mathbb{E}_{q_t}[g] - 2\mathbb{E}_{p_{\lambda_t}^*}[g])(\eta \mathbb{E}_{q_t}[g])$$

$$= - \left\| \mathbf{r}_{q_t, p_{\lambda_t}^*} \right\|_{\mathcal{F}_t}^2 + \eta(\mathbb{E}_{q_t}[g] - \mathbb{E}_{p_\lambda^*}[g])^2 - \eta(\mathbb{E}_{p_{\lambda_t}^*}[g])^2$$

$$\leq - \left\| \mathbf{r}_{q_t, p_{\lambda_t}^*} \right\|_{\mathcal{F}_t}^2 + \eta(\mathbb{E}_{q_t}[g] - \mathbb{E}_{p_\lambda^*}[g])^2 - \eta(\frac{1}{2}(\mathbb{E}_{q_t}[g])^2 - (\mathbb{E}_{q_t}[g] - \mathbb{E}_{p_\lambda^*}[g])^2)$$

$$\leq - \left\| \mathbf{r}_{q_t, p_{\lambda_t}^*} \right\|_{\mathcal{F}_t}^2 + 2\eta(\mathbb{E}_{q_t}[g] - \mathbb{E}_{p_\lambda^*}[g])^2 - \frac{1}{2}\eta(\mathbb{E}_{q_t}[g])^2$$

$$\leq -(1 - 2c_1\eta) \left\| \mathbf{r}_{q_t, p_{\lambda_t}^*} \right\|_{\mathcal{F}_t}^2 - \frac{1}{2}\eta(\mathbb{E}_{q_t}[g])^2$$

$$\leq -(1 - 2c_1\eta) \mathbb{D}_{\mathcal{F}_t}(q_t, p_{\lambda_t}^*)^2 - \frac{1}{2}\eta(\mathbb{E}_{q_t}[g])^2.$$

If $\lambda_t = 0$ and $\mathbb{E}_{q_t}[g] \leq 0$, we have

$$\frac{\mathrm{d}}{\mathrm{d}t} E(q_t, \lambda_t) = - \left\| \mathbf{r}_{q_t, p_0^*} \right\|_{\mathcal{F}_t}^2 = -\mathbb{D}_{\mathcal{F}_t}(q_t, p_0^*)^2$$

So we can check that $\frac{d}{dt}\mathbb{E}(q_t, \lambda_t) \leq -\Delta(q_t, \lambda_t)$ in both cases. Combing the two cases yield the result. Therefore,

$$\min_{t \in [0, T]} \Delta(q_t, \lambda_t) \leq \frac{1}{T} \int_0^T \Delta(q_t, \lambda_t)\mathrm{d}t \leq \frac{1}{T}(E(q_0, \lambda_0) - E(q_T, \lambda_T)) \leq \frac{1}{T} E(q_0, \lambda_0).$$

$\square$

*Proof of Theorem 2.6.*  First note that

$$L(q, \lambda^*) = \mathrm{KL}(q \,||\, p_{\lambda^*}^*) + \Phi(\lambda^*).$$

So if $q^*$ is the solution to problem 13, it is of form $p_{\lambda^*}^*$.

For simplicity, we denote $G(q) = \mathbb{E}_{x \sim q}[g(x)]$. Recall the dual objective function is:

$$\Phi(\lambda) = \min_q L(q, \lambda) = -\log \mathbb{E}_{x \sim p_0^*}[\exp(-\lambda g(x))].$$

Its derivatives are given by

$$\dot{\Phi}(\lambda) = \frac{\mathbb{E}_{x \sim p_0^*}[g(x) \exp(-\lambda g(x))]}{\mathbb{E}_{x \sim p_0^*}[\exp(-\lambda g(x))]} = \mathbb{E}_{x \sim p_\lambda^*}[g(x)] = G(p_\lambda^*),$$

$$\ddot{\Phi}(\lambda) = -\frac{\mathbb{E}_{x \sim p_0^*}[g(x)^2 \exp(-\lambda g(x))]}{\mathbb{E}_{x \sim p_0^*}[\exp(-\lambda g(x))]} + \left( \frac{\mathbb{E}_{x \sim p_0^*}[g(x) \exp(-\lambda g(x))]}{\mathbb{E}_{x \sim \pi}[\exp(-\lambda g(x))]} \right)^2$$

$$= -\mathrm{var}_{p_\lambda^*} g(x).$$

Therefore Assumption 2.2 indicates $\Phi$ is $v_0$-strongly-concave and $v_1$-smooth. As a consequence,

$$\frac{1}{2} v_0 (\lambda^* - \lambda)^2 \le \Phi(\lambda^*) - \Phi(\lambda) \le \frac{2}{v_0} \dot{\Phi}(\lambda)^2 = \frac{2}{v_0} G(p_{\lambda_t}^*)^2. \tag{22}$$

Next, we check the evolution of $E(q_t, \lambda_t)$. If $\lambda_t \ge 0$ or $\mathbb{E}_{q_t}[g] \ge 0$,

$$\frac{\mathrm{d}}{\mathrm{d}t} E(q_t, \lambda_t) = - \left\| \mathbf{r}_{q_t, p_0^*} - \lambda_t \mathbf{s}_{q_t, g} \right\|_{\mathcal{F}_t}^2 + \eta (\mathbb{E}_{q_t}[g] - \mathbb{E}_{p_{\lambda_t}^*}[g])^2 - \eta (\mathbb{E}_{p_{\lambda_t}^*}[g])^2$$

$$\le -(1 - c_1 \eta) \left\| \mathbf{r}_{q_t, p_0^*} - \lambda_t \mathbf{s}_{q_t, g} \right\|_{\mathcal{F}_t}^2 - \eta (\mathbb{E}_{p_{\lambda_t}^*}[g])^2$$

$$(\text{Using } (18),(22)) \le - \min\{(1 - c_1 \eta)/\kappa, \tfrac{1}{2}\eta v_0\}(\mathrm{KL}(q_t, p_{\lambda_t}^*) + \Phi(\lambda^*) - \Phi(\lambda_t))$$

$$= - \min\{(1 - c_1 \eta)/\kappa, \tfrac{1}{2}\eta v_0\} E(q_t, \lambda_t).$$

If $\lambda_t = 0$ and $\mathbb{E}_{q_t}[g] < 0$, recall that $G(p_0^*) > 0$, so

$$\frac{\mathrm{d}}{\mathrm{d}t} E(q_t, \lambda_t) = - \left\| \mathbf{r}_{q_t, p_0^*} - \lambda_t \mathbf{s}_{q_t, g} \right\|_{\mathcal{F}_t}^2$$

$$\le -(1 - c_1 \eta) \left\| \mathbf{r}_{q_t, p_0^*} - \lambda_t \mathbf{s}_{q_t, g} \right\|_{\mathcal{F}_t}^2 - \eta (G(p_{\lambda_t}^*) - G(q_t))^2$$

$$\le -(1 - c_1 \eta) \left\| \mathbf{r}_{q_t, p_0^*} + \lambda_t \mathbf{s}_{q_t, g} \right\|_{\mathcal{F}_t}^2 - \eta (G(p_{\lambda_t}^*))^2$$

$$(\text{Using } (18),(22)) \le - \min\{(1 - c_1 \eta)/\kappa, \tfrac{1}{2}\eta v_0\}(\mathrm{KL}(q_t, p_{\lambda_t}^*) + \Phi(\lambda^*) - \Phi(\lambda_t))$$

$$= - \min\{(1 - c_1 \eta)/\kappa, \tfrac{1}{2}\eta v_0\} E(q_t, \lambda_t).$$

In conclusion, we find the following always hold

$$\frac{\mathrm{d}}{\mathrm{d}t} E(q_t, \lambda_t) \le - \min\{(1 - c_1 \eta)/\kappa, \tfrac{1}{2}\eta v_0\} E(q_t, \lambda_t).$$

This leads to linear convergence of $E(q_t, \lambda_t)$.

Finally we note that

$$E(q_t, \lambda_t) = \mathrm{KL}(q_t \| p_{\lambda^*}^*) + (\Phi(\lambda^*) - \Phi(\lambda)).$$

So by the Young's inequality and that

$$(\lambda - \lambda^*)(G(p_\lambda^*) - G(p_{\lambda^*}^*)) = (\lambda - \lambda^*)(\dot{\Phi}(p_\lambda^*) - \dot{\Phi}(p_{\lambda^*}^*)) \le 0,$$

we have

$$\mathrm{KL}(q_t \| p_{\lambda^*}^*) = \mathrm{KL}(q_t \| p_\lambda^*) + (\lambda^* - \lambda)G(q_t) + \Phi(\lambda) - \Phi(\lambda^*)$$

$$= \mathrm{KL}(q_t \| p_\lambda^*) + (\lambda^* - \lambda)(G(q_t) - G(p_\lambda^*)) + (\lambda^* - \lambda)(G(p_\lambda^*) - G(p_{\lambda^*}^*)) + \Phi(\lambda) - \Phi(\lambda^*)$$

$$\le \mathrm{KL}(q_t \| p_\lambda^*) + \frac{1}{2}(G(q_t) - G(p_\lambda^*))^2 + \frac{1}{2}(\lambda^* - \lambda)^2 + (\lambda^* - \lambda)(\dot{\Phi}(\lambda) - \dot{\Phi}(\lambda^*)) + \Phi(\lambda) - \Phi(\lambda^*)$$

$$\le (1 + \frac{1}{2}c_3)\mathrm{KL}(q_t \| p_\lambda^*) + \frac{1}{2}(1 + v_1)(\lambda^* - \lambda)^2$$

$$\le (1 + \frac{1}{2}c_3)\mathrm{KL}(q_t \| p_\lambda^*) + \frac{1}{2v_0}(1 + v_1)(\lambda^* - \lambda)^2(\Phi(\lambda) - \Phi(\lambda^*))$$

$$\le \max\{1 + \frac{1}{2}c_3, \frac{v_1}{2v_0} + \frac{1}{2v_0}\} E(q_t, \lambda_t).$$

$\square$

## B.2 Proofs for the constraint controlled method

For simplicity, we denote

$$\lambda'_t = \frac{\alpha \mathbb{E}_{q_t}[g] + \langle \mathbf{r}_{q_t, p_0^*}, \, \mathbf{s}_{q_t, g} \rangle_{\mathcal{F}_t}}{\|\mathbf{s}_{q_t, g}\|^2_{\mathcal{F}_t}},$$

and $G(q) = \mathbb{E}_{\theta \sim q}[g(\theta)]$. We also note that the following hold

$$\mathbf{r}_{q, p_0^*} - \lambda \mathbf{s}_{q, g} = \mathbf{r}_{q, p_\lambda^*}$$

because $\mathbf{R}_{q, p_0^*} - \lambda \mathbf{S}_{q, g} = \mathbf{R}_{q, p_\lambda^*}$.

**Lemma B.2.** *Let $\mathcal{F}$ be a Hilbert space and assume the linear operator $\mathbf{R}(\phi) := \mathbb{E}_q[(\nabla \log p - \nabla \log q)\phi]$ and $\mathbf{S}(\phi) := \mathbb{E}_q[\nabla_\theta g^\top \phi]$ yield a Riesz representation $\mathbf{r}_p$ and $\mathbf{s}$, that is, $\mathbf{R}(\phi) = \langle \mathbf{r}_p, \phi \rangle_{\mathcal{F}}$ and $\mathbf{S}(\phi) = \langle \mathbf{s}, \phi \rangle_{\mathcal{F}}$, then the optimal solution of*

$$\min_{\phi \in \mathcal{F}} -\langle \mathbf{r}_p, \, \phi \rangle_{\mathcal{F}} + \frac{1}{2} \|\phi\|^2_{\mathcal{F}} \quad s.t. \quad \langle \mathbf{s}, \, \phi \rangle_{\mathcal{F}_t} \leq -\alpha \mathbb{E}_{q_t}[g],$$

*is*

$$\phi^* = \mathbf{r}_p - \lambda^* \mathbf{s} = \mathbf{r}_{p_\lambda^*}$$

*where $p_\lambda \propto p(x) \exp(-\lambda g)$ and*

$$\lambda^* = \max \left( \frac{\alpha \mathbb{E}_q[g] + \langle \mathbf{r}_p, \, \mathbf{s} \rangle_{\mathcal{F}}}{\|\mathbf{s}\|^2_{\mathcal{F}}}, \, 0 \right)$$

*Proof.* Introducing a Lagrange multiplier $\lambda \geq 0$. Since the problem is convex in $\phi$, we can try to solve the dual of the problem, which is

$$\max_{\lambda \geq 0} \min_{\phi \in \mathcal{F}} -\langle \mathbf{r}_p, \, \phi \rangle_{\mathcal{F}} + \frac{1}{2} \|\phi\|^2_{\mathcal{F}} + \lambda \langle \mathbf{s}, \phi \rangle_{\mathcal{F}} + \lambda \alpha \mathbb{E}_q[g]$$

$$= \max_{\lambda \geq 0} \min_{\phi \in \mathcal{F}} \langle \tfrac{1}{2}\phi + \lambda \mathbf{s} - \mathbf{r}_p, \, \phi \rangle_{\mathcal{F}} + \lambda \alpha \mathbb{E}_q[g].$$

Given $\lambda$, the optimal $\phi$ is obtained by

$$\phi^* = \mathbf{r}_p - \lambda \mathbf{s}.$$

Note that

$$\langle \phi^*, \psi \rangle_{\mathcal{F}} = \mathbb{E}_q[(\nabla \log p - \nabla \log q - \lambda \nabla g)\psi] = \mathbb{E}_q[(\nabla \log p_\lambda - \nabla \log q)\psi],$$

so $\phi^* = \mathbf{r}_{p_\lambda}$. Plug this solution back to the dual problem, we find the dual problem is given by

$$\max_{\lambda \geq 0} -\frac{1}{2} \|\mathbf{r}_p - \lambda \mathbf{s}\|^2_{\mathcal{F}} + \lambda \alpha \mathbb{E}_q[g].$$

Since this problem is quadratic in $\lambda$, we find can $\lambda^*$ as claimed. $\square$

*Proof of Theorem 2.7.* Note that if $\lambda'_t \geq 0$

$$\langle \mathbf{r}_{q_t, p_{\lambda_t}^*}, \mathbf{s}_{q_t, g} \rangle_{\mathcal{F}_t} = \langle \mathbf{r}_{q_t, p_0^*}, \mathbf{s}_{q_t, g} \rangle_{\mathcal{F}_t} - \lambda_t \|\mathbf{s}_{q_t, g}\|^2_{\mathcal{F}_t}$$

$$= \langle \mathbf{r}_{q_t, p_0^*}, \mathbf{s}_{q_t, g} \rangle_{\mathcal{F}_t} - \alpha G(q) - \langle \mathbf{r}_{q_t, p_0^*}, \mathbf{s}_{q_t, g} \rangle_{\mathcal{F}_t} = -\alpha G(q). \qquad (23)$$

If $\lambda'_t \leq 0$, $\lambda_t = 0$ and

$$\langle \mathbf{r}_{q_t, p_{\lambda_t}^*}, \mathbf{s}_{q_t, g} \rangle_{\mathcal{F}_t} = \langle \mathbf{r}_{q_t, p_{\lambda_t'}^*}, \mathbf{s}_{q_t, g} \rangle_{\mathcal{F}_t} + \lambda'_t \|\mathbf{s}_{q_t, g}\|^2_{\mathcal{F}_t}$$

$$= -\alpha G(q) + \lambda'_t \|\mathbf{s}_{q_t, g}\|^2_{\mathcal{F}_t}. \qquad (24)$$

Therefore

$$\frac{d}{dt} G(q_t) = \langle \mathbf{s}_{q_t, g}, \phi_t \rangle_{\mathcal{F}_t} = \langle \mathbf{s}_{q_t, g}, \mathbf{r}_{q_t, p_{\lambda_t}^*} \rangle_{\mathcal{F}_t}$$

$$= \begin{cases} -\alpha G(q), & \lambda'_t > 0 \\ -\alpha G(q) + \lambda'_t \mathbb{E}\|s_{q_t, g}\|^2_q, & \lambda'_t \leq 0 \end{cases}.$$

Note that in either case
$$\frac{d}{dt}G(q_t) \le -\alpha G(q_t).$$
This leads to the first claim using the Gronwall's inequality.

Next, we note the followings either $\lambda'_t > 0$ or $\lambda'_t \le 0$,
$$\lambda_t \langle \mathbf{s}_{q_t,g}, \mathbf{r}_{q_t,p^*_{\lambda_t}} \rangle_{\mathcal{F}_t} = -\lambda_t G(q).$$

This leads to
$$\begin{aligned}
\frac{d}{dt}\mathrm{KL}(q_t \,||\, p^*_0) &= -\langle \mathbf{r}_{q_t,p^*_0}, \phi_t \rangle_{\mathcal{F}_t} \\
&= -\|\mathbf{r}_{q_t,p^*_{\lambda_t}}\|^2_{\mathcal{F}_t} - \lambda_t \langle \mathbf{r}_{q_t,p^*_{\lambda_t}}, \mathbf{s}_{q_t,g} \rangle_{\mathcal{F}_t} \\
&= -\|\mathbf{r}_{q_t,p^*_{\lambda_t}}\|^2_{\mathcal{F}_t} + \lambda_t \alpha G(q).
\end{aligned}$$

This leads to our second claim, since $\lambda_t \ge 0$ and $\|\mathbf{r}_{q_t,p^*_{\lambda_t}}\|^2_{\mathcal{F}_t} \ge 0$. $\qquad\square$

*Proof of Theorem 2.9.* Following on our derivation in Theorem 2.7, we find that
$$\begin{aligned}
\mathbb{D}_{\mathcal{F}_t}(q_t, \, p^*_{\lambda_t})^2 &= \left\| \mathbf{r}_{q_t,p^*_0} - \lambda_t \mathbf{s}_{q_t,g} \right\|^2_{\mathcal{F}_t} \\
&= -\frac{d}{dt}\mathrm{KL}(q_t \,||\, p^*_0) + \lambda_t \alpha \mathbb{E}_{q_t}[g].
\end{aligned}$$

Therefore,
$$\int_0^T \mathbb{D}_{\mathcal{F}_t}(q_t, \, p^*_{\lambda_t})^2 + (-\lambda_t \alpha \mathbb{E}_{q_t}[g])_+ dt = \mathrm{KL}(q_0 \,||\, p^*_0) - \mathrm{KL}(q_T \,||\, p^*_0) + \alpha \int_0^T (\lambda_t \mathbb{E}_{q_t}[g])_+ dt.$$

Note that by Theorem 2.7
$$\alpha \int_0^T (\lambda_t \mathbb{E}_{q_t}[g])_+ dt \le \alpha \lambda_{\max,+} \int_0^T \exp(-\alpha t)(\mathbb{E}_{q_0}[g])_+ dt = \lambda_{\max,+}(\mathbb{E}_{q_0}[g])_+.$$

The result then follows. $\qquad\square$

*Proof of Theorem 2.6.* First note that
$$L(q, \lambda^*) = \mathrm{KL}(q \,||\, p^*_{\lambda^*}) + \Phi(\lambda^*).$$
So if $q^*$ is the solution to problem 13, it is of form $p^*_{\lambda^*}$.

To show linear convergence, we investigate the KL divergence. When $\lambda'_t \ge 0$,
$$\begin{aligned}
\frac{d}{dt}\mathrm{KL}(q_t \,||\, p^*_{\lambda^*}) &= -\langle \mathbf{r}_{q_t,p^*_{\lambda_t}}, \mathbf{r}_{q_t,p^*_{\lambda^*}} \rangle_{\mathcal{F}_t} \\
&= -\|\mathbf{r}_{q_t,p^*_{\lambda_t}}\|^2_{\mathcal{F}_t} + (\lambda_* - \lambda_t)\langle \mathbf{r}_{q_t,p^*_{\lambda_t}}, \mathbf{s}_{q_t,g} \rangle_{\mathcal{F}_t} \\
&= -D_{\mathcal{F}_t}(q_t, p^*_{\lambda_t})^2 + \alpha(\lambda_t - \lambda_*)G(q).
\end{aligned}$$

When $\lambda'_t < 0$, using $\lambda_t = 0$ and (24)
$$\begin{aligned}
\frac{d}{dt}\mathrm{KL}(q_t \,||\, p^*_{\lambda^*}) &= -\|\mathbf{r}_{q_t,p^*_{\lambda_t}}\|^2_{\mathcal{F}_t} + (\lambda_* - \lambda_t)\langle \mathbf{r}_{q_t,p^*_{\lambda_t}}, \mathbf{s}_{q_t,g} \rangle_{\mathcal{F}_t} \\
&= -\|\mathbf{r}_{q_t,p^*_{\lambda_t}}\|^2_{\mathcal{F}_t} + (\lambda_t - \lambda_*)(\alpha G(q) - \lambda'_t \|\mathbf{s}_{q_t,g}\|^2_{\mathcal{F}_t}) \\
&= -\|\mathbf{r}_{q_t,p^*_{\lambda_t}}\|^2_{\mathcal{F}_t} - \alpha \lambda_* G(q) + \lambda_* \lambda'_t \|\mathbf{s}_{q_t,g}\|^2_{\mathcal{F}_t} \\
&\le -\|\mathbf{r}_{q_t,p^*_{\lambda_t}}\|^2_{\mathcal{F}_t} + \alpha(\lambda_t - \lambda_*)G(q).
\end{aligned}$$

In other words, in both cases, we have
$$\frac{d}{dt}\mathrm{KL}(q_t \,||\, p^*_{\lambda^*}) \le -\|\mathbf{r}_{q_t,p^*_{\lambda_t}}\|^2_{\mathcal{F}_t} + \alpha(\lambda_t - \lambda_*)G(q).$$

Next we use condition (18),

$$\|\mathbf{r}_{q_t, p^*_{\lambda_t}}\|^2_{\mathcal{F}_t} = D_{\mathcal{F}_t}(q_t, p^*_{\lambda_t})$$
$$\geq \frac{1}{c_2} \text{KL}(q_t \| p^*_{\lambda_t})$$
$$\geq \alpha \text{KL}(q_t \| p^*_{\lambda_t})$$
$$= \alpha \int q_t(x) \log \frac{q_t(x)}{p^*_{\lambda_t}(x)} dx$$
$$= \alpha \int q_t(x) \log \frac{q_t(x)}{\frac{1}{Z_{\lambda_t}} q^*(x) \exp(-(\lambda_t - \lambda_*)g(x))} dx$$
$$= \alpha \int q_t(x) \left( \log \frac{q_t(x)}{p^*_{\lambda^*}(x)} + (\lambda_t - \lambda_*)g(x) + \log Z_{\lambda_t} \right) dx$$
$$= \alpha(\text{KL}(q_t \| p^*_{\lambda^*}) + (\lambda_t - \lambda_*)G(q_t) + \log Z_{\lambda_t})$$
$$\geq \alpha(\text{KL}(q_t \| p^*_{\lambda^*}) + (\lambda_t - \lambda_*)G(q_t)),$$

where we have used that $\log Z_\lambda = -\Phi(\lambda) \geq -\Phi(\lambda^*) = 0$. Plug this inequality into our derivation of KL divergence,

$$\frac{d}{dt}\text{KL}(q_t \| p^*_{\lambda^*}) \leq -\alpha\text{KL}(q_t \| p^*_{\lambda^*}) - \alpha(\lambda_t - \lambda_*)G(q_t) + \alpha(\lambda_t - \lambda_*)G(q_t)$$
$$= -\alpha\text{KL}(q_t \| p^*_{\lambda^*}).$$

By Gronwall's inequality, we find that

$$\text{KL}(q_t \| p^*_{\lambda^*}) \leq e^{-\alpha t}\text{KL}(q_0 \| p^*_{\lambda^*}).$$

$\square$

## C  More Details on the Experiments: Settings and Results

### C.1  Results on Gaussian Mixtures

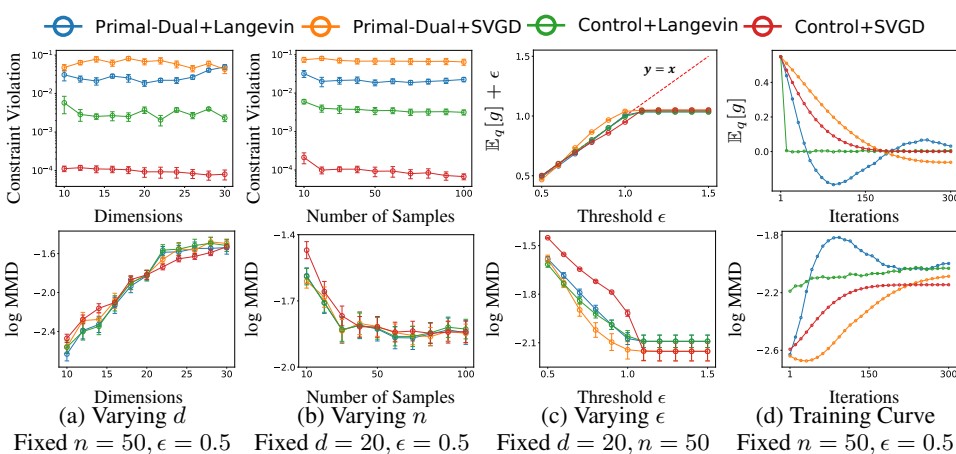

(a) Varying $d$    (b) Varying $n$    (c) Varying $\epsilon$    (d) Training Curve
Fixed $n = 50, \epsilon = 0.5$   Fixed $d = 20, \epsilon = 0.5$   Fixed $d = 20, n = 50$   Fixed $n = 50, \epsilon = 0.5$

Figure 4: Results on Gaussian mixture models. Averaged over 10 runs

In this experiment, We set $p^*_0$ to be randomly generated $\mathbb{R}^d$-Gaussian mixtures: $p^*_0(\theta) = \frac{1}{m}\sum_{i=1}^m \mathcal{N}(\theta; \mu_i, \sigma_i^2)$ where $m$ is fixed to 5 in all the experiments. $\mu_i$ and $\sigma_i$ are $d$-dimensional vectors, which are randomly sampled from $\mathcal{N}(0, I)$ and Uniform$(0, I)$, respectively. We set the constraint to be $g(\theta) = \|\theta\|^2 - \epsilon$, which constrains the second order moment of the samples.

We run 300 iteration for all the four methods. In all the experiments, we perform grid search on the hyper-parameters for each method, and choose the result with smallest absolute constraint error. We 50 particles unless mentioned otherwise.

Figure 4 shows the result as we vary the dimension $d$ of the domain (Figure 4(a)), the number of particles $n$ (Figure 4(b)), the threshold value $\epsilon$ (Figure 4(c)), and iteration step (Figure 4(d)). The violation of constraint is measured by $(\mathbb{E}_q[g])_+$; the optimization result is measured by the maximum mean discrepancy, $\mathrm{MMD}(q, p_0^*)$, between $q$ and $p_0^*$, because the KL divergence $\mathrm{KL}(q \,\|\, p_0^*)$ is not computable when $q$ is an empirical measure of particles. We can see that all the methods achieve similar $\mathrm{MMD}(q, p_0^*)$, but `Control+SVGD` tends to satisfy the constraint best on (a) and (b).

We provide in Figure 5 the optimization curve with different step size $\epsilon$ for `Primal-Dual` methods.

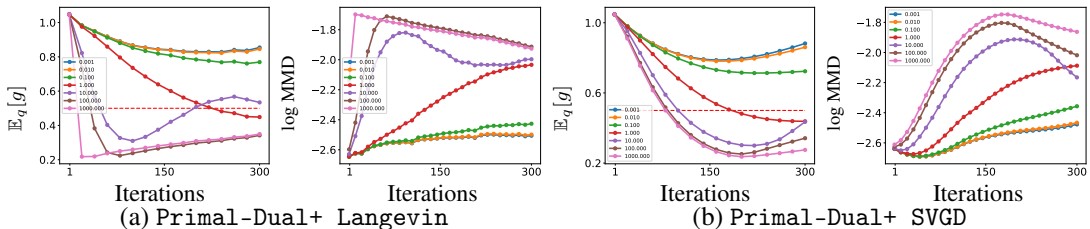

Figure 5: The change of $\mathrm{MMD}(q_t, p_0^*)$ and $\mathbb{E}_{q_t}[g]$ vs. iteration $t$ on the Gaussian mixture model. The colors represent different different step size $\epsilon$.

## C.2    Settings of Experiments in the Main Paper

**Bayesian Logistic Regression with Logic Rules**      In this experiment, we consider Bayesian logistic regression for binary classification. The problem of interest is to predict whether or not to lend loans to a specific applicant. The dataset, lending club loan data [2], contains complete loan data for all loans issued through 2007-2015 of several banks. Each data point is a 28-dimensional feature including the current loan status, latest payment information, and other additional features. We use 50 particles. Here, we define the logic loss as the binary cross-entropy loss.

**Monotonic Bayesian Neural Networks**      In this experiment, we use the COMPAS dataset (J. Angwin & Kirchner, 2016). COMPAS is a dataset containing the criminal records of 6,172 individuals arrested in Florida. The task is to predict whether the individual will commit a crime again in 2 years. The probability predicted by the system will be used as a risk score. We use 13 attributes for prediction. The risk score should be monotonically increasing w.r.t. four attributes, `number of prior adult convictions`, `number of juvenile felony`, `number of juvenile misdemeanor`, and `number of other convictions`. The bayesian neural network is built up upon a two-layer ReLU neural network with 100 hidden neurons. We use 10 particles to sample from the posterior.

**Fair Bayesian Neural Networks**      In this experiment, we adopt the setting in (Martinez et al., 2020; Liu & Vicente, 2020). The experiment is performed on the *Adult Income* dataset (Kohavi, 1996), which contains 30,162 training samples and 15,060 test samples. It is a binary classification problem, whose prediction target is whether the income of a person is higher than 50,000 dollars per year. Following (Martinez et al., 2020; Liu & Vicente, 2020), we randomly sample a subset of 20,000 data points from the training set as our training set. Each data point has a 86-dimensional feature. We also follow (Liu & Wang, 2016) to use two-layer neural network with ReLU activation. The network has 50 hidden neurons, containing 4401 parameters in total. We use 50 particles to sample from the posterior.

## C.3    Additional Results

**Additional Training Plots**    We provide more plots on training fair Bayesian neural networks here, including the training LL, constraint loss and the change of $\lambda_t$ vs. iterations. See Fig. 6 and Fig. 7.

**Unconstrained Baselines**    We provide the results of unconstrained baselines in this section for comparison with the constrained methods in the main text. These points are too far away from the constrained ones so we did not put them in the figures. See Tab. 1 for the results. Unconstrained

---

[2]https://www.kaggle.com/wendykan/lending-club-loan-data

| Bayesian Logistic Regression with Logic Rules | | | | |
|---|---|---|---|---|
| Methods | Training LL | Logic Loss | Test Accuracy | Test Violation |
| SVGD | -11.83 ± 0.6 | 1.42 ± 0.55 | 0.649 ± 0.005 | 0.996 ± 0.099 |
| Langevin | -12.53 ± 0.3 | 1.43 ± 0.49 | 0.648 ± 0.003 | 0.998 ± 0.087 |

| Training Monotonic Bayesian Neural Networks | | | | |
|---|---|---|---|---|
| Methods | Training LL | Mono Loss | Test LL | Test Mono Loss |
| SVGD | -0.842 ± 0.022 | 0.156 ± 0.21 | -0.665 ± 0.017 | 0.013 ± 0.006 |
| Langevin | -0.864 ± 0.105 | 0.213 ± 0.23 | -0.666 ± 0.031 | 0.016 ± 0.008 |

| Training Fair Bayesian Neural Networks | | | | |
|---|---|---|---|---|
| Methods | Training LL | Fairness Loss | Test Accuracy | Disparate Impact |
| SVGD | -0.512 ± 0.015 | 0.214 ± 0.009 | 0.849 ± 0.013 | 0.185 ± 0.021 |
| Langevin | -0.525 ± 0.024 | 0.148 ± 0.015 | 0.847 ± 0.005 | 0.184 ± 0.011 |

Table 1: Experiment results on unconstrained baselines.

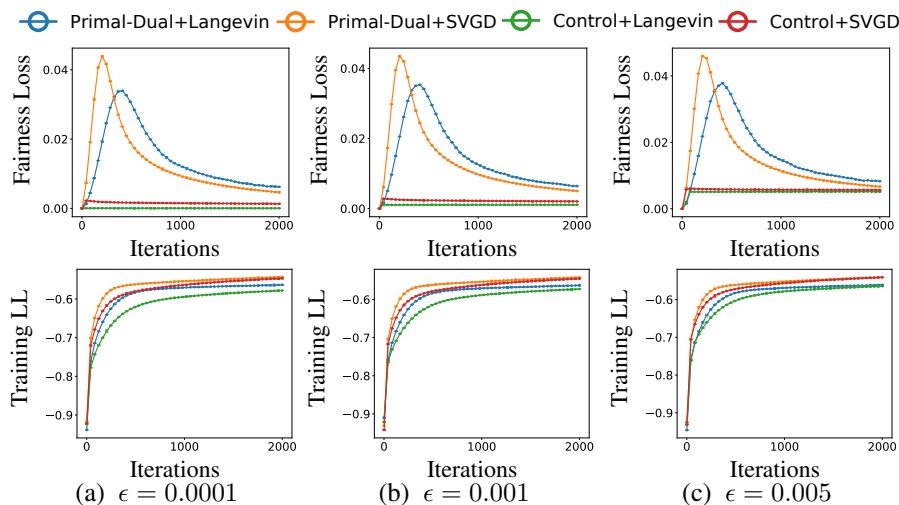

Figure 6: Experiment results on learning fair Bayesian neural networks. 'LL': log-likelihood.

baselines typically has higher training LL and much higher loss on the constraints, compared with our proposed constrained sampling methods.

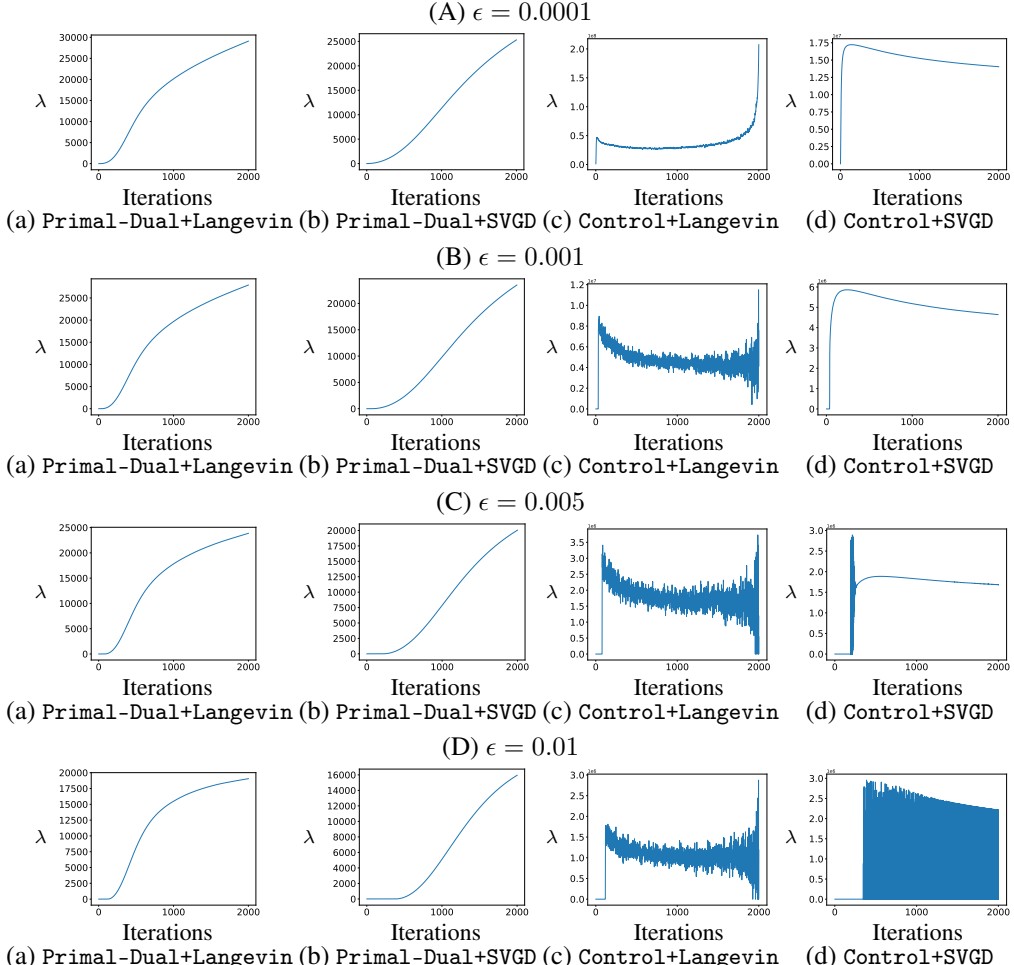

Figure 7: Plot of $\lambda$ vs. iterations on learning fair Bayesian neural networks.

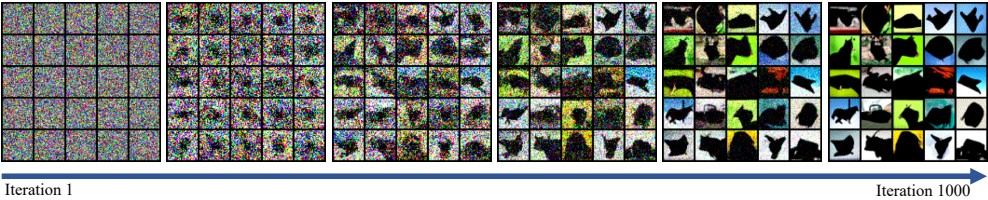

Figure 8: Intermediate results of `Control+Langevin` with NCSN