# OpenReview forum: "Sampling  with Trusthworthy Constraints:  A Variational Gradient Framework   "
_NeurIPS.cc/2021/Conference — NeurIPS 2021 Poster_

### Official Review · Reviewer_mFWB · 2021-07-14

**Rating:** 6
**Confidence:** 4

**Summary:**

The authors propose two ways to extend Unadjusted Langevin Algorithm (ULA) and Stein Variational Gradient Descent (SVGD) for sampling from a distribution known up to its normalization constant, under moment constraints. The first one is a primal-dual gradient descent scheme,
while the second one is a controlled gradient descent one.

They provide convergence guarantees in continuous time for both schemes: a 1/T rate for a given optimality criterion Delta in the more generic case (Th 3.4 and 3.8), and linear convergence rate in KL if some form of log-concavity is satisfied for the current target p_{lamba_t}* (Th 3.5 and 3.9).

These methods perform reasonably well on synthetic and real world experiments, with a better performance for controlled-based methods.


**Ethics Review Area:**

["I don’t know"]

**Limitations And Societal Impact:**

Limitations : cf comments, e.g. regarding theoretical guarantees and SVGD.

**Main Review:**

The paper tackles the interesting problem of sampling under moment constraints. I am quite convinced about the importance of this problem, as motivated in the introduction or the experiments conducted (e.g. bayesian logistic regression with logic rules, or training fair bayesian neural networks).

They proposed two extensions of ULA and SVGD which are widely used in Bayesian inference, based on schemes from the optimization literature.
These ideas are natural but are novel up in sampling to the best of my knowledge. Also, the authors did a good job in providing both theoretical guarantees and empirical evidence for the relevance of these schemes for sampling.

My main concern is regarding the theoretical guarantees.
Regardig the first scheme, assumption 3.1 is problematic, as it has to be satisfied at all times for the current target distribution p_lambda_t *.
For instance, eq 14 corresponds to assuming the log-Sobolev inequality for p_lambda_t* when F_t=L2(q_t)/in the Langevin setting (satisfied for instance when p_lambda_t* is c_2 strongly log concave) as explained by the authors, or to the Stein log Sobolev inequality when F_t= RKHS /in the SVGD setting, which is hardly satisfied (see Lemma 36 in “on the geometry of Stein variational gradient descent” , Duncan, Nusken, 2019 - does not hold for a bounded kernel such as gaussian or IMQ if p_lambda_t* has exponential tails).

But maybe it is common in the euclidean optimization framework to require such assumptions on the current iterates to prove convergence for primal-dual gradient methods. Can the authors point to such references?I have a similar concern for the second scheme, and the requirement that the norm of s_{q_t, g} is non zero or lower bounded by c_ .

Still, Theorem 3.5  and 3.9 are mainly valid for the Langevin setting, but are not likely to be satisfied for SVGD -> this should be written clearly.

Overall, I tend towards a weak accept.


Minor comments:

Section 2:

When F_t is a RKHS, D is known to metrize convergence between q_t and p_0* only under specific conditions on p_0* and the reproducing kernel, see “Measuring sample quality” Gorham and Mackey, 2017. In particular it may fail for a gaussian kernel, which is widely used in practice. This is worth mentioning in the paper.

l101: whose time-discretisation -> one of the time discretisation leads to Unadjusted Langevin Monte Carlo algorithm. Many other time-discretizations of this flow exist, see  “Sampling as optimisation over the space of measures”, A. Wibisono, Colt 2018; or “Wasserstein Proximal Gradient” A. Salim, A. Korba, G. Luise, Neurips 2020.

Section 3:

eq 10: what is eta?
eq 11 and 12: p*-> po*?
l189 : grad log q -> grad loq po*
l196: same with p

Assumption 3.7 is not written entirely…
Figure 1: I would have been curious about the non-constrained baseline (ie classical langevin or svgd) to have a better idea of the attained loss and E_q[g(\theta)]

Section 4 :

l256 : i have troubles understanding how the distribution D_logic is created. For the points x that do not satisfy the logic constraint, it looks like this constraint penalizes the original KL objective as well; but maybe i’m getting it wrong.

What is the fairness loss and disparate impact? Why don’t you plot the (average) prediction loss of the BNN and the constraint violation as in the first experiments?


**Time Spent Reviewing:**

5

---

> ### Author Response · Authors · 2021-08-10
> **Authors' Response to Reviewer mFWB**
>
> Thank you a lot for your time and comments!
>
>
> **1. Regarding Assumption 3.1**
>
> For the Langevin case, the condition is satisfied for all $\lambda_t\geq0$ if  both $g$ and $f$ to be strongly convex, since $f+\lambda_t g$ will also be strongly convex (Note also that $\lambda_t$ is always non-negative since it is a the Lagrangian multiplier). This was briefly discussed at line 142 in the original manuscript. There is no known result for the SVGD case, which we have clarified and cited [DNS19] in the revision.
>
> [DNS19] Duncan, A., Nuesken, N. & Szpruch, L. On the geometry of Stein variational gradient descent. arXiv.org, arXiv:1912.00894 [stat.ML], (2019).
>
> **2. Are the results apply to SVGD? ("Still, Theorem 3.5 and 3.9 are mainly valid for the Langevin setting, but are not likely to be satisfied for SVGD -> this should be written clearly.")**
>
> First of all, our writing might has confused you into thinking  the results do not apply to SVGD. LSI type condition is only required to obtain exponential convergence in KL divergence in SVGD (see Theorem 3.5 and 3.9); it **DOES NOT** require LSI to obtain the $1/T$ convergence in Stein discrepancy as we show in Theorem 3.4 and 3.8. We realize this confusion might be because we have put LSI with other milder conditions together in Assumption 3.1. In the revision, we have split it into a single assumption 3.2. We have also added in [DNS19] to discuss this assumption and the restriction.
>
> In the finite dimensional optimization literature (see e.g., "Lectures on Convex Optimization" by Y. Nesterov),
> it is well known  that gradient descent type of algorithms require strong convexity to have linear rate. Otherwise, with just convexity, one can show convergence with rate 1/T. Our results fit quite well with this dichotomy.
>
> Our requirement that $s_{q_t, g}\neq 0$ is similar to requiring the gradient is non-zero for gradient descent. It is easy to see that if the gradient is zero at a non-optimal point, applying gradient descent type of algorithm can stuck there.
> We have a similar discussion after Assumption 3.8.
>
> **3. Problem of Gaussian kernel ("In particular it may fail for a gaussian kernel, which is widely used in practice.")**
>
> Thank you for bringing up this point. We have added this point in Section 2 after we discussed SVGD.
>
> **4. Typos & Additional References**
>
> Thank you. We have fixed the typos and added the references as you suggested.
>
> **5. Unconstrained Baselines**
>
> We provide the unconstrained baselines in Appendix C. Please refer to table 1 in the Appendix.
>
> **6. Regarding $D_{logic}$ in L256**
>
> We are sorry about the confusion. $\Omega_{logic}$ is the region of $(x,y)$ such that (1) an applicant $x$ has the lowest credit rank and not employed, but $y$ is accepted, or (2) an applicant $x$  has the highest credit rank and has been employed over 15 years, but $y$ is rejected. We consider a uniform distribution $D_{logic}$ over $\Omega_{logic}$.
> The loss function is defined as $\ell_{logic}(\theta)=E_{(x, y)\sim D_{logic}}\mathcal{L}\left[1 - y, \hat{y}(x;\theta)\right]$, where $\mathcal{L}$ is the standard logistic regression loss. So we penalize the model if it violates the logic rules.
>
> **7. What is the fairness loss and disparate impact?**
>
> Disparate impact is a metric of fairness in machine learning, which can be measured by the CV score,
> $\text{CV}(f) = |p(\hat{y}=1|A=0) - p(\hat{y}=0|A=0)|$,
> where $f$ is our classifier, $\hat{y}$ is the prediction, and $A$ is the sensitive attribute (for example, gender). Usually, $\hat{y}=1$ if $f(x)>0$; otherwise, $\hat{y} =0$. CV score implies the gap between the probabilities of getting positive outcomes in different sensitive groups (for exmaple, between males and females). Our fairness loss is a convex surrogate of CV score,
> $\ell_2(f) =  \left[ E_{x_i, a_i \in \mathcal{D}}(a_i-\bar{a})f(x_i) \right]^2$, where $x_i$ is the feature of instance $i$; $a_i$ is the sensitive attribute of instance $i$; $\mathcal{D}$ is the dataset, and $\bar{a}$ is the averaged value of all $a_i$ in the dataset. Basically, it is minimizing the empirical covariance between the the prediction and the sensitive attribute.
>
> **8. Why don’t you plot the (average) prediction loss of the BNN and the constraint violation as in the first experiments?**
>
> Average prediction loss for BNN is in fact the training log-likelihood, which we reported in our figures. Constraint violation ($(E_q[g])_+$) is the absolute value between the obtained constraint loss and our expected threshold $\epsilon$. For BNN experiments, our figures report the obtained constraint loss (fariness loss, logic loss and monotonicity loss), which is $E_q[g]+\epsilon$ like column $(c)$ in Figure 1.

---

> > ### Comment · Reviewer_mFWB · 2021-08-21
> > **After rebutal**
> >
> > Sorry for the late answer. Thanks for your answers that were clarifying. Some comments below.
> >
> > 1) regarding assumption 3.1
> > Thank you for clarifying, I agree with your argument in the Langevin/Log sobolev case. Would be great additionally if you could provide examples where g is convex.
> > For SVGD/Stein log sobolev however, this assumption is a bit problematic.
> >
> > 2) regarding th 3.4, 3.5, 3.8, 3.9
> > Maybe there was a misunderstanding, but I had no problems with the theorems 3.4 and 3.8 in both the Langevin and SVGD case.
> > My concern was on th 3.5 and 3.9 (exponential convergence in KL) in the svgd case. As said in my rebutal, I think these results should be stated only in the Langevin case, or it should be very clear that they are unlikely to hold for SVGD, at least in the setting of the impossibility result of Duncan et al. (ie, the target as exponential tails - as for a gaussian for instance - and the kernel grows at most polynomially - which basically includes all the practical kernels, from polynomials, gaussian, laplace-like kernels).
> >
> > Minor comment : Also, there might have been a misunderstanding on my question regarding assumptions.
> > I agree that in classical optimization, convexity of the function to be optimized is needed to get linear rates.
> > In sampling/Langevin, this can be translated in  (1) assuming a log-sobolev assumption on the target p_0.
> > However, this is different from (2) assuming a  log-sobolev assumption on the iterates p_{\lambda_t} - which is stronger. The authors have clarified that (1) implies (2) if g convex, which seems more reasonable as g is fixed.
> > My question was more a curious one, I was wondering if in classical primal-dual optimization algorithms , there were assumptions on the trajectory/the iterates - which is different from assuming something on the fixed objects such as the function to be optimized.

---

> > > ### Author Response · Authors · 2021-08-28
> > > **Thanks!**
> > >
> > > Dear Reviewer, thanks a lot for your efforts and your reply. As we said in the earlier rebuttal, we will clarify the issue of SLI condition in the SVGD case.
> > >
> > > Regrading your question on assumptions on trajectories vs. fixed objectives, we think the analysis of the classical (finite dimensional) primal-dual gradient method can be done using the same approach as our method, and hence there will be similar assumptions on the trajectories which in turn can be implied by an assumption on the fixed objective and constraints. This is probably true in general since the analysis is fundamentally about understanding properties of the trajectory, while we hope the condition can be reduced to the fixed objective/constraints, so that it can be easily verified. Are we interpreting your questions correctly?
> > >
> > > Regarding the example when $g$ is convex, there are certainly many practical problems when the constraint function $g$ is convex (e.g., the norm constraints $g(x) = \norm{x}^2$). But are you asking for something else?

---

### Official Review · Reviewer_1Fbx · 2021-07-15

**Rating:** 6
**Confidence:** 3

**Summary:**

This paper proposes to modify existing approximation posterior inference algorithms to satisfy an additional moment constraint that might be related to trustworthiness, fairness, or something else. Two algorithms are proposed, which can be used with either a discretized Langevin algorithm or Stein variational gradient descent. Convergence properties of the underlying gradient flows are also characterized.

**Limitations And Societal Impact:**

These are appropriately addressed

**Main Review:**

Originality: Overall the work seems quite original and creative. The moment constraint is introduced by extending the usual KL minimizing gradient flow problem as a minimax problem. This leads to a rather elegant “primal-dual” gradient flow formulation, which is discretized to obtain the first proposed algorithm. The second method is obtained as a trust-region–type method using a linearization of the constraint. In addition to proposing two algorithms, convergence results for the (continuous-time, infinite-particle limit) of both algorithms are provided.

Quality: The paper appears to be technically sound. However, the results concerning the SVGD flow are misleading. This is because most of the theory requires a log Sobolev inequality to hold. However, it appears that an LSI essentially never holds for SVGD [DNS19]. Thus, the results as they apply to SVGD are essentially vacuous. Moreover, we should not expect SVGD to behave well for multimodal distributions, since the KSD does a poor job detecting the relative mass of different mods. Given all this, it is unsurprising that the Langevin algorithms are empirically superior. But the superiority of the Langevin algorithms is never clearly acknowledged. If anything, the author(s) seem favorably disposed to the SVGD approach, despite its clear theoretical and empirical inferiority.

Clarity: While the motivation of the paper is clear, the theoretical results are extremely dense and hard to digest – particularly for a reader who isn’t already familiar with the relevant background material on gradient flows. In light of the better performance of the Langevin-based algorithms, to improve readability I would suggest moving all of the technical SVGD content and the general framework to the appendix and focus on the more digestible Langevin case.

Significance: The results and approaches developed seem to be potentially quite useful and could open up a number of interesting new avenues for future work. Thus, I’m inclined to accept the paper. Though I do strongly encourage the author(s) to revise the paper as suggested above. I think making the paper easier to read will also make it more impactful as it will reach a broader audience.


[DNS19] Duncan, A., Nuesken, N. & Szpruch, L. On the geometry of Stein variational gradient descent. arXiv.org, arXiv:1912.00894 [stat.ML], (2019).

**Time Spent Reviewing:**

1.5

---

> ### Author Response · Authors · 2021-08-10
> **Authors' Response to Reviewer 1Fbx**
>
> Thank you a lot for your time and comments!
>
> **1. Empirical performance of SVGD vs. Langevin variants**
>
> In our original submission, Langevin variants tends to have higher log-likelihood than SVGD in Figure 2 and Figure 3. We have investigated this issue and found that it is due to a sub-optimal (and incomparable) choices of step size, rather than the inherent properties of the different methods:  We used a decaying step size for Langevin dynamics, and an Adagrad step size for SVGD. It turns out that the Adagrad step size tends to yield lower performance, which is why we saw lower likelihood on SVGD. We have updated experiments to allow better comparison of the methods, which we describe as follows:
>
> In our old implementation, we follow the optimization strategy in the original papers of SVGD and SGLD: Adagrad for SVGDs and decaying step size ($\epsilon_t = \epsilon_0 (b+t)^{-\gamma}, \gamma=0.55, b=1.0$, as suggested in [1]) for SGLD. We perform additional experiments on the choice of optimization strategy on learning fair Bayesian neural networks, and present the results in the following figure:
> <https://drive.google.com/file/d/1gSVPKfZsXCqMlZcthCweHlJqA4S51iR9/view?usp=sharing>
>
> In the new experiments, we  test three step size choices for both SVGD and Langevin methods: 1) constant step size $\epsilon_t = \epsilon_0$, 2) decaying step size $\epsilon_t  = \epsilon_0 (b+t)^{-\gamma}$ where $\gamma = 0.55$ and $b=1$, and 3) the Adagrad schedule (whose master step size $\epsilon_0$ and the other parameters is set as default in PyTorch). For each algorithm, we search the best $\epsilon_0$ in the grid $[1e-3, 1e-4, 1e-5, 1e-6]$ that achieves the lowest constraint loss at the end of the training.  We observe that, when using the same step size schedule, the SVGD  variants in fact tend to have higher training log-likelihood than the Langevin variants. And both variants tend to perform best when using the decaying step size. In particular, because we used adagrad for SVGD and decaying step size for Langevin in our original submission, we saw Langevin performs better as a result of using better step size schedule.
>
> [1] Welling M, Teh Y W. Bayesian learning via stochastic gradient Langevin dynamics[C]//Proceedings of the 28th international conference on machine learning (ICML-11). 2011: 681-688.
>
> **2. Theoretical comparison of SVGD vs. Langevin**
>
> We want to clarify some issues on the theoretical properties of SVGD vs. Langevin dynamics.
>
> 1. Our writing might has confused you. LSI type condition is only required to obtain the exponential convergence in KL divergence in SVGD (see Theorem 3.5 and 3.9); it **DOES NOT** require LSI to obtain $1/T$ convergence in Stein discrepancy as we show in Theorem 3.4 and 3.8. We realize this confusion might be because we have put LSI with other milder conditions together in Assumption 3.1. In the revision, we have split it into a single assumption 3.2. We have also added in [DNS19] to discuss this assumption.
>
> 2.  For SVGD, what is needed for exponential convergence is that $KL(q_t, p)\leq \alpha KSD^2(q_t, p),~\forall t\in[0,\infty)$  where $q_t$ is the density at time $t$ in the SVGD trajectory (denote it by Condition I); this is weaker than the standard LSI like assumption, which would require that  $KL(q, p)\leq \alpha KSD^2(q, p)$ holds for any probability measure $q$ (Condition II). It is clear that (Condition II) is false for KSD since if we choose $q$ to be a discrete measure, we would have $KL(q, p) =\infty$ but $KSD(q,p) < \infty$ (assuming the kernel and the target distribution is sufficiently regular). On the other hand, because the actual SVGD trajectory $\{q_t\colon t\in [0,\infty)\}$ have more regularity (which depends on the regularity of  initialization), we think we can not exclude the possibility that (Condition I) may still hold under proper conditions (although it is certainly a stronger condition if it exits); but this remains an open question.
>
> 3. Note that the current analysis only considers the continuous-time and infinite-particle limit (i.e., $n\to\infty$), and hence they are not a direct indication of practical efficiency,  because we need  to factor in the time and particle discretization in terms of which SVGD and Langevin are very different: Langevin algorithm uses a random diffusion process while SVGD uses a deterministic interacting particle system. They have their own pros and cons depending on the evaluation metric. For example, SVGD would be clearly better in terms of practical performance if we are restricted to using $n=1$ particle (e.g.,  due to memory constraints) since one-particle SVGD reduces to maximizing the posterior distribution (i.e., MAP). In general,  Langevin algorithm attempts to approximate the whole distribution with diffusion and hence requires to average on many particles, across many iterations (after the burn-in period), while SVGD is better suited with very small number of particles and can be terminated with less iterations as soon as the KSD is close to zero because it is a deterministic process. SVGD also has a very different quadrature approximation interpretation which works in the small $n$ region. (see "Liu, Wang, Stein Variational Gradient Descent as Moment Matching, NeurIPS 18").
>
> **3. Removing the SVGD part?**
>
> We agree the Langevin algorithm is more well-known, and focusing on that may simplify our presentation in Section 2, where we review the two algorithms jointly. However, these two algorithms are two very different approaches and have their own comparative advantages as we mentioned earlier. Langevin has been widely known  and well-understand, but there is also a  growing interest in SVGD as a new method. The idea of this is also tightly integrated with our unified view of these two methods, which remove of which would largely deviate from our original motivation.  We hope to attract audience from both sides and show that our framework applies to both approaches.
> And given the new empirical results that show that advantages of SVGD over Langevin, we think keeping the unified treatment is the best option.
>
>   We will polish the work to easy the readers who are interested in only one method, by for example, indicating in the beginning of Section 2 that "Our results apply to both algorithms. For audience with special interest in one of the two algorithms, we recommend ignoring the other one in the first read." We have went through the manuscript to make sure this claim is correct.

---

> > ### Comment · Reviewer_1Fbx · 2021-08-17
> > **thank you**
> >
> > Thank you for clarifying key points related to when an LSI is needed and the experiments. With the revised experiments showing similar performance for the SVGD and Langevin algorithms, I think it is appropriate to retain the general treatment. But please do include some discussion of why the LSI assumption appears to be quite strong in the SVGD case.

---

> > > ### Author Response · Authors · 2021-08-28
> > > **Thanks!**
> > >
> > > Dear Reviewer, thanks a lot for your efforts in reviewing the draft. We will clarify the issue of LSI and SVGD.

---

### Official Review · Reviewer_HwFo · 2021-07-16

**Rating:** 6
**Confidence:** 4

**Summary:**

The paper presents SVGD variants that can handle constraints on the optimized distribution in the form of an expectation under it. The methods are based on Lagrangian multiplier (primal-dual form) and constraint controlled optimization, respectively. Their convergence analyses in the context of a constraint are developed. Experiment verifies a good fit to the target distribution with the satisfaction of the constraint.

**Limitations And Societal Impact:**

Addressed.

**Main Review:**

Under a critical view, the considered task seems not to require fancy techniques. The primal-dual form transforms the target distribution to satisfy the constraint, and the constraint control version enforces the constraint to hold in every optimization step (similar to projection in each step). So I did not feel lighted on its originality. Nevertheless, the paper has done this in some fine details, including convergence analyses that take the constraint into consideration, and a good introduction and application results on tasks that require such a constraint.

One technical question: The authors mentioned that the conditions of CCGF is milder than the primal-dual version. Does that mean that CCGF indeed has a better convergence guarantee in more scenarios, or is it just a proving technique issue?

Although the organization is logical and intuitive explanations of math are provided, the writing of the paper can be further improved. To name a few:
  - Line 2, "to handling". Line 21, "which frame". Three lines above Line 106, "to be an". Line 112, missing "(". Line 144, "holds". Line 153, "$E$" but not "\mathbb{E}". Line 166, "direction". Second line in Page 6, "in addition". Line 174, "allows us to have". Line 202, "defined in (17), we". Line 225, "trustworthy".
  - Line 130: "The empirical distribution of ... is used to approximate ... in (19)": I suppose the samples are $\{\theta_i\}$ but not $\{\xi_i\}$, and Eq. (9) but not (19).
  - In Eq. (10), what is $\eta$? A step size parameter is not expected in such a continuous-time equation.
  - Eq. (14): $D(q_t, p^*_{\lambda_t})$ but not $D(q_t \Vert p^*_{\lambda_t})$.

**Time Spent Reviewing:**

2

---

> ### Author Response · Authors · 2021-08-10
> **Authors' Response to Reviewer HwFo**
>
> Thank you a lot for your time and comments!
>
> **1. Comparing CCGF and primal-dual methods**
>
> These two methods are derived from two different perspectives. CCGF tries to fit the constraints directly, while the primal-dual operates in a more indirect manner. This is reflected in the proof techniques, as we can directly show the constraint decays linearly for the CCGF method, but we can only control the constrain through a Lyapunov function with the primal dual method. We believe that the conditions required by CCGF is intrinsically weaker than that of primal-dual, but we do not have a formal proof of it since it is entangled with the different proof techniques. This was an issue that we thought about when preparing the draft but did not elaborate it in the draft.  We really appreciate that you pointed out this issue  and will modify the draft to clarify it.
>
> **2. Typos**
>
> Thank you for careful reading. We have corrected the typos you mentioned.
>
> **3. In Eq. (10), what is $\eta$?**
>
> $\eta$ is the  velocity ratio between $\lambda_t$ updates and  the $q_t$ updates. A larger $\eta$ leads to faster updates of $\lambda_t$ compared to updates of $q_t$. It is necessary when in the continuous time limit.

---

> > ### Comment · Reviewer_HwFo · 2021-08-22
> > **Thanks for Authors' Response**
> >
> > Thanks for your reply. Hope there could be a discussion on the relations between the primal-dual and CCGF methods (e.g., which one has a better theoretical guarantee or requires a milder condition, and if the merit comes at a cost). Also, the authors should improve the presentation of the paper throughout. I just found many additional typos and symbol misuses in the second read.

---

> > > ### Author Response · Authors · 2021-08-28
> > > **Thanks!**
> > >
> > > Dear reviewer, thank you for your reply. As we said in the rebuttal, we will extend the discussion on primal-dual vs. CCGFs and polish the draft. Among others, we think one of the key practical (and inherently theoretical) draw back of the primal-dual method is that its performance depends on the initialization of $\lambda_t$ and the step size $\eta$. Following Theorem 3.4, the maximum step size $\eta$ is determined by the constant $1/c_1$ in Eq (13), which may be very small or even zero depending on the choice of $f$ and $g$ (especially when they are non-convex). But the CCGF method does not have this issue. See also line 214-215 of the original draft in which we briefly remarked the additional conditions needed for the primal-dual method.

---

### Official Review · Reviewer_ZxYc · 2021-07-16

**Rating:** 8
**Confidence:** 3

**Summary:**

The authors extend gradient flow methods for variational inference to handle inequality constraints. These inequality constraints can capture fairness, safety, interpretability, and other such desiderata. They introduce two methods to solve this. First, they introduce a 'primal-dual gradient method': they formulate the equivalent minimax problem to the constrained optimization, and then they derive the dynamics for gradient descent ascent of the zero-sum game. Convergence is shown by explicitly providing a Lyapunov function. Next, they introduce a 'constraint controlled gradient descent', which first finds the feasible set, and then stays within the feasible set afterward (inspired by controlled barrier functions (CBFs) from control). Convergence is shown similarly to the work in CBFs.

**Ethical Concerns:**

This paper raises no ethical concerns, and seeks to address ethical concerns introduced by the deployment of machine learning methods.

**Limitations And Societal Impact:**

The work provides algorithms to include constraints in variational inference, and provides theoretical guarantees. However, as stated by the authors, the theoretical analysis primarily considers the mean-field limit. Additionally, the theorems are for continuous-time convergence, and the conditions for the discretizations to follow the continuous-time flow are not provided. Nevertheless, this paper provides a significant contribution in its own right, and these limitations are more good directions for future work rather than a shortcoming of the current paper.

Additionally, it's unclear how practical these algorithms would be in real-world settings, as the simulations provided are primarily small-scale. However, a deeper evaluation of this would be a paper in its own right, and is outside the scope of this paper, whose primary goal is theoretical development.


**Main Review:**

Originality: This work introduces the problem of variational inference subject to inequality constraints. The tasks are new. Whereas the methods are not new, they are combined in a novel fashion to solve a new, well-motivated problem.

Quality: The methods used are appropriate for the problem at hand, and the theoretical claims are (mostly) technically sound. However, one alarming issue is the fact that Assumption 3.7 is missing from the draft (it abruptly ends before the assumption is given). This MUST be fixed. However, the methods used are relatively standard and the results are intuitive; again, the main contribution of this paper is the new problem formulation.

Clarity: The paper is well-written; it provides a good overview of the methods, both formally and intuitively. It provides intuition for how the results are derived, as well as full rigorous proofs in the appendix.

Significance: The problem introduced is definitely of significance to the community and society at large.

Major issues:
Assumption 3.7 is incomplete, ending with "...we have". This MUST be addressed.

Minor notes (did not affect evaluation):
- Typo in the title: 'Trusthworthy'
- Line 52: 'contorl'
- Equation 3: $\hat y(\theta; \theta)$.
- Line 86: $r_q$ should be bolded for consistency.
- Line 108: I recommend the authors describe the median trick in greater detail.
- Line 180: 'qudratic'
- Line 224: 'Gaussian mixture toy' -> 'Gaussian mixture toy model'
- Line 225: 'thrustworthy'
- Line 285: 'manifold' -> 'manifolds'


**Time Spent Reviewing:**

3

---

> ### Author Response · Authors · 2021-08-10
> **Authors' Response to Reviewer ZxYc**
>
> Thank you a lot for your time and comments!
>
> **1. Regarding novelty of the method**
>
> We were motivated to pursue the current work because the space of **constrained**
> functional variational inference
> (in the Langevin+SVGD framework)
> with complex non-linear constraints has been basically empty and we have not seen algorithms of similar kind (especially the constrained controlled version) in the literature (please kindly point out any reference that we may miss).
>
> We should also point out that the connection with CBFs only serves as a remote motivation. The methods, problem settings, theories in the CBF literature, which is largely on the control problems in robotics, is very different [1].
> Also, our analysis,  the constraint controlled version  in particular, is not a trivial practice;   we have not seen similar works in the literature.
>
> [1] Ames et.al., Control Barrier Functions: Theory and Applications
>
> **2. Assumption 3.7 is incomplete**
>
> We are sorry that part of  Assumption 3.7 was missing due to typographical error.
> We have fixed this issue.The complete assumption is:
>
> " There exists an upper bound $\lambda_{\max, +} < \infty$, such that
> for all time $t$ when the constraint is not satisfied (i.e., $E_{q_t}[g] >0$), we have
>  $\lambda_t \leq \lambda_{\max, +}$ in Eq. (17)."
>
> **3. More description of median trick**
>
> The median trick is a standard approach for selecting the bandwidth $h$ of the kernel functions (e.g., RBF, $k(x,y)=\exp(-||x-y||^2/h^2)$) in  the kernel method literature.
> Assume we have $n$ particles $\{x_i\}$. It sets the bandwidth to be
> $h = \mathbf{Median}\{||x_i-x_j||\colon i\neq j\}$. This allows the scale of the kernel to be adapted with the data (while not being dominantly impacted by outliers).
>
> **4. Typos**
>
> Thanks for pointing out these typos, we have fixed all of them.
>
> **5. Regarding real-world settings**
>
> We agree that it would be appealing to include more larger-scale experiments, which we would like to investigate in future works. We should also point out that the computational overhead that we add compared with the unconstrained SVGD and Langevin dynamics is very minimum and hence we are not very inferior in terms of scalability compared with the unconstrained versions; and the scale of our current experiments  are in par  with that from typical Bayesian inference papers these days. We agree that further scaling up these methods are of independent interest with great potential impact.

---

### Author Response · Authors · 2021-08-10
**Authors' General Comment**

We thank the reviewer for the valuable feedback. We have revised the draft based on your comments and will make further improvement in the final draft. See below our response to your individual questions. Please let us know if you have any further questions.

---

### Decision · Program_Chairs · 2021-09-27

**Decision:**

Accept (Poster)

**Comment:**

The reviewers agreed that this paper should be accepted. For the camera ready: aside from generally going through the reviews/responses and ensuring that all suggested changes are made, the reviewers had one more remark for the authors. In particular, while the reviewers appreciated the effort to compare Langevin/SVGD, they still strongly suggest that the authors suitably nuance the limitations of the SVGD theory. One idea brought forward by a reviewer that the others thought was acceptable is to state results for Langevin and defer SVGD to the appendix with careful statements of limitations. This highlights that they can be obtained in a very similar manner, but that they are less likely to hold in practice.